# PROACTIVEBENCH: BENCHMARKING PROACTIVENESS IN MULTIMODAL LARGE LANGUAGE MODELS

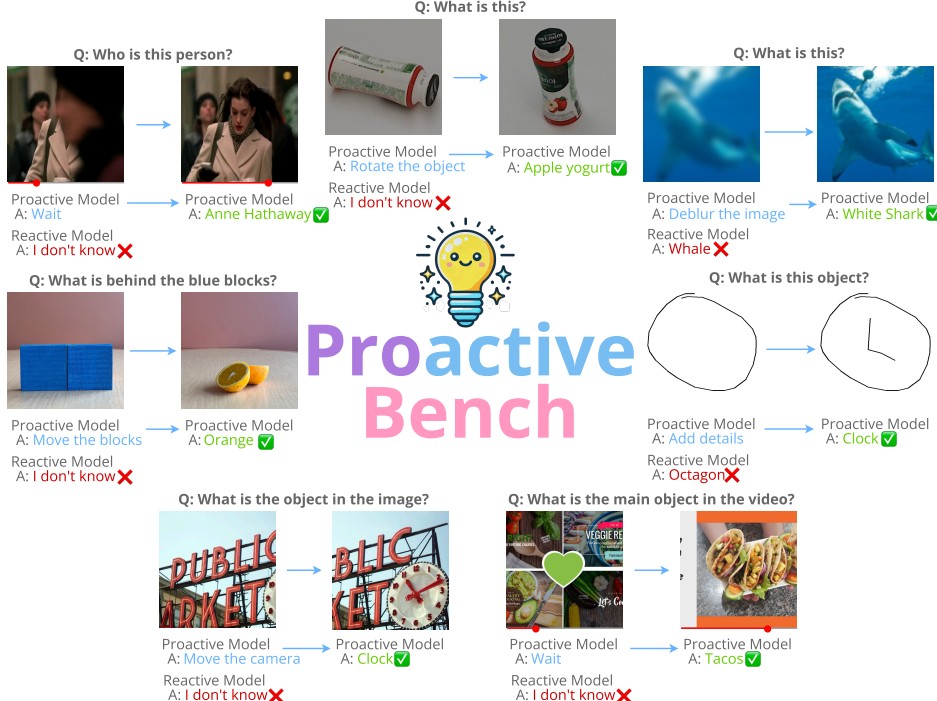

Figure 1: We propose **ProactiveBench**, a multimodal benchmark to evaluate *proactiveness* in multimodal large language models, i.e., the ability to ask for additional visual cues from the user to answer an ambiguous query. ProactiveBench tests proactiveness in seven scenarios involving partially observable objects and individuals, blurred input, and temporally evolving scenes.

## ABSTRACT

How do multimodal large language models (MLLMs) handle images where the object of interest is partially or fully occluded? While a human would naturally ask follow-up questions or seek additional visual cues before answering, do MLLMs exhibit similar "proactive" behavior by prompting the user for more information? Despite their growing use in collaborative settings, no benchmark currently evaluates the proactiveness of MLLMs. To fill this gap, we introduce ProactiveBench, a benchmark built from seven repurposed datasets to evaluate proactiveness across tasks such as recognizing occluded objects, enhancing image quality, and interpreting coarse sketches, to name a few. We evaluated 21 MLLMs on ProactiveBench and found that they generally lack proactiveness. Model capacity shows no clear correlation with proactiveness, and adding "hints" in the query to elicit proactive suggestions yields only marginal gains. Surprisingly, conversation histories and in-context learning introduce negative biases, hindering performance. Overall, our results highlight the challenge of instilling proactiveness in MLLMs, with ProactiveBench being a first step toward building more proactive models.

# 1 INTRODUCTION

Studies in neuroscience suggest that meaningful perception of the world arises from dynamic interaction with our environment (Goodale & Milner, 1992; Haskins et al., 2020; Shapiro, 2007; Heuer et al., 2020). Faced with incomplete or ambiguous information, we instinctively generate hypotheses, proactively search for additional clues, and revise our interpretations.

This ongoing cycle of inquiry and refinement is currently unexplored for multimodal large language models (MLLMs) (Zhu et al., 2025; Li et al., 2025; Bai et al., 2025), where ambiguities may arise when a user's query is unanswerable due to false user premises (Wu et al., 2024) or bad image quality (Chiu et al., 2020). For instance, for the query "What is behind the blue blocks?" of Fig. 1, a model can answer directly, e.g., hallucinating an incorrect answer (Li et al., 2023b), or abstaining (Whitehead et al., 2022; Guo et al., 2024). Such behavior is called *reactive*. Conversely, a more desirable response is to ask the user for additional visual cues, e.g., by moving the blocks to reveal the hidden object. We refer to such behavior as *proactive*, since it aims to refine predictions by asking the user to intervene, providing additional information. With the wide adoption of MLLMs, an essential question arises: can they, like humans, proactively seek for additional visual cues?

To fill this gap we introduce ProactiveBench, a novel benchmark to evaluate MLLMs' proactiveness in multiple scenarios, by repurposing seven existing datasets (ROD (Lee et al., 2023), VSOD (Liao et al., 2020), MVP-N (Wang et al., 2022a), ImageNet-C (Hendrycks & Dietterich, 2019), Quick-Draw (Jongejan et al., 2016), ChangeIt (Souček et al., 2022), and MS-COCO (Lin et al., 2014)) with different target tasks (e.g., sketch recognition, product identification) that require user intervention to answer correctly. As Fig. 1 shows, ProactiveBench captures different aspects of proactiveness: (temporal) occlusion removal, camera movement, object movement, image quality enhancement, and asking for details. In total, it contains more than 108k images grouped into 18k samples featuring 19 proactive suggestions. Each sample (see Fig. 2) contains the starting ambiguous frame, the reference frame with complete information, and all the frames in between. The user intervention results in a new frame with more visual cues based on the model's guidance (termed *proactive suggestion*).

We tested 21 state-of-the-art MLLMs (e.g., LLaVA-OV (Li et al., 2025), Qwen2.5-VL (Bai et al., 2025), InternVL3 (Zhu et al., 2025)) on ProactiveBench, reporting accuracy and number of proposed proactive suggestions before predicting the category. Our experiments suggest that evaluated models lack proactiveness, i.e., are reactive. Thus, they either tend to abstain from answering (saying, e.g., "I don't know") or predict random categories when the visual cues are insufficient, as Fig. 1 shows. Providing hints about proactive suggestions increases their sampling probability, which marginally raises accuracy. Interestingly, while some MLLMs (e.g., LLaVA-NeXT Vicuna 7B, InternVL3 1B) appear on the surface as more proactive than others (e.g., LLaVA-OV 7B, Qwen2.5-VL 7B, InternVL3 8B), via a controlled experiment we show that the higher proactiveness results from a lower rate of abstention on unanswerable questions, rather than a deeper understanding of the problem. Instead, conditioning on the conversation history or few-shot samples increases proactiveness but reduces accuracy. Finally, our results highlight that proactiveness is not an emerging property in MLLMs and must be explicitly elicited, showcasing the challenging nature of ProactiveBench.

**Contributions**: (i) We formalize and explore MLLMs proactiveness in a wide spectrum, promoting the development of models that can ask user assistance under uncertainty; (ii) We introduce ProactiveBench, a novel open-source benchmark that assesses MLLM's proactiveness in diverse contexts; (iii) Our evaluation of 21 MLLMs on ProactiveBench reveals limited proactiveness of current models, even when explicitly hinting it, highlighting the challenges of this setting.

# 2 RELATED WORK

**MLLMs.** Earlier MLLMs emerged from pioneering efforts to extend frozen LLMs, such as Frozen (Tsimpoukelli et al., 2021) and Flamingo (Alayrac et al., 2022). These seminal works convert pre-trained LLMs by injecting visual tokens in the language model's attention layers and fine-tuning them. Subsequent models, like PaLI (Chen et al., 2022), BLIP (Li et al., 2022; 2023a), LLaVA (Liu et al., 2023a; 2024a), and InstructBLIP (Dai et al., 2023), simplified the architecture by forwarding projected visual tokens as input to the LLM and reducing the parameter count. Furthermore, LLaVA (Liu et al., 2023a) proposes fine-tuning LLMs using instruction tuning data, improving

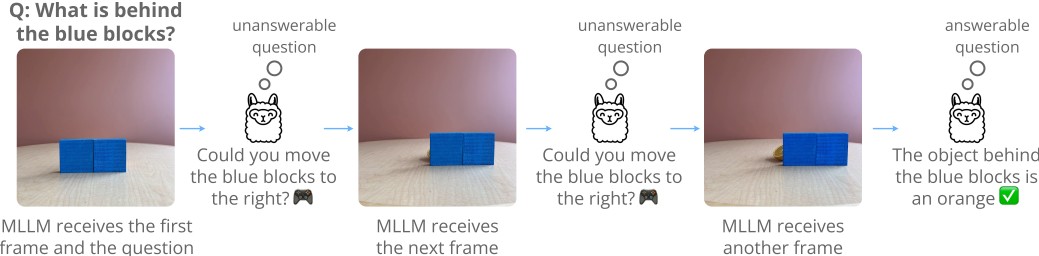

Figure 2: **ProactiveBench evaluation.** At step 1, the MLLM should propose to move the occluding object (proactive suggestion), as the question is unanswerable. ProactiveBench, then, returns a new frame following the MLLM's suggestion. Since the model is still unsure, it asks to move the blocks again. Finally, step 3 holds sufficient information, allowing the MLLM to predict the answer.

data efficiency and reasoning capabilities. We focus on benchmarking the proactive capabilities of such models on a broad spectrum of tasks, a previously unexplored research direction.

**Benchmarking for MLLMs.** While early efforts evaluate MLLMs on visual question answering (Antol et al., 2015; Goyal et al., 2017; Marino et al., 2019), a second wave of benchmarks focused on tasks requiring reasoning and world knowledge (Liu et al., 2024d; Li et al., 2023b; Liu et al., 2024c; Yue et al., 2024; Kazemi et al., 2023). As recent MLLMs support multiple images and videos as inputs, more complex, multi-input benchmarks have been introduced to evaluate their reasoning capabilities (Kil et al., 2024; Kazemi et al., 2024; Dingjie et al., 2024; Fu et al., 2024; Meng et al., 2024; Wang et al., 2024; Tong et al., 2024; Jiang et al., 2024; Li et al., 2024a). A parallel effort has emerged in the embodied AI literature, where several studies evaluate agents that integrate LLMs (Li et al., 2024b; Shridhar et al., 2020; Padmakumar et al., 2022; Wang et al., 2022b; Savva et al., 2019). However, none of these works benchmark MLLMs' proactiveness to ambiguous or even unanswerable queries. Related to our work, Liu et al. (2024b) explores whether MLLM's directional guidance can help visually impaired individuals in capturing images. However, Liu et al. (2024b) limits the evaluation to a single type of proactive suggestion and to single-turn conversations, not measuring the effectiveness of the MLLM's proposed suggestion. Instead, we investigate models' proactiveness in seven distinct scenarios over multiple turns, enabling a more comprehensive analysis of failure cases and false proactive behaviors.

**Active vision** improves perception (Aloimonos et al., 1988) by allowing an active observer to control sensing strategies (e.g., viewpoint) dynamically. Active vision has been extensively studied in view planning (i.e., determining optimal sensor viewpoints) (Zeng et al., 2020), object recognition (Browatzki et al., 2012), scene and 3D shape reconstruction (Smith et al., 2021), and robotic manipulation (Chuang et al., 2024). To overcome passive systems' drawbacks, Xu et al. (2023) introduces an open-world *synthetic* game environment, where agents actively explore their surroundings, performing multi-round abductive reasoning. Although we inherit the underlying spirit of active vision, our work differs as: (i) ProactiveBench contains real-world images from diverse and complex scenarios, (ii) the observer receives feedback from the MLLM, through proactive suggestions, fostering a collaboration of the model and the user, ideal for human-machine cooperative tasks.

## 3 THE PROACTIVEBENCH

This section introduces ProactiveBench, detailing the evaluation of MLLM proactiveness (Sec. 3.1), the repurposed datasets used (Sec. 3.2), and a filtering pipeline ensuring questions require MLLMs to ask for human intervention (Sec. 3.3). Model and dataset licenses are provided in Appendix E.

### 3.1 EVALUATING PROACTIVENESS IN MLLMS

We study MLLMs' *proactiveness*, where a model should either answer correctly or suggest how to make a question answerable. Since suggestions may leave questions unresolved (e.g., Fig. 2's central frame), we evaluate proactiveness in a multi-turn setting, allowing the MLLM to interact with the environment over multiple steps. Evaluating models in this setting requires verifying the

answer and applying the requested action to get a new state. This can be challenging in free-form generation, as (i) answers must be parsed by an LLM, adding cost and computational overhead, and (ii) proposed actions may be inapplicable. Therefore, we focus on the multiple-choice question-answering framework, where models select the answer from multiple options, enabling structured evaluation over multiple turns. For completeness, we report results in free-form generation in Appendix A.

We follow previous works (Duan et al., 2024; Liu et al., 2023b) and frame the evaluation as a Markov decision process $(\mathcal{S}, \mathcal{A}, \pi_\theta, \mathcal{R})$, over a finite states space $\mathcal{S}$, a discrete set of actions $\mathcal{A}$, a policy $\pi_\theta$ (the MLLM), and reward $\mathcal{R}$. At step $t$, the model observes state $s_t \in \mathcal{S}$, which comprises the image $\mathcal{I}_t$ and valid actions $\mathcal{A}_t \subseteq \mathcal{A}$. The model selects an action $a_t$ conditioned by the question $q$ (e.g., "what is this object?") and the state $s_t = \{\mathcal{I}_t, \mathcal{A}_t\}$. Thus, the transition function $\mathcal{T} : \mathcal{S} \times \mathcal{A} \to \mathcal{S}$ is defined by the conditioned policy $\pi_\theta(a_t|q, s_t)$. By selecting a proactive suggestion (e.g., "move the occluding object"), state $s_t$ transitions to $s_{t+1}$, leading to a new image and a new set of valid actions. Instead, by either abstaining (e.g., "I do not know") or selecting a wrong category (e.g., dog vs. cat), the evaluation stops with a wrong prediction. As environments are discrete, the policy can select proactive suggestions a finite number of times, depending on the datasets, after which the evaluation terminates with a wrong prediction. Finally, the evaluation also terminates if the model predicts the correct answer. For each MLLM, we report the average accuracy and the average number of proactive suggestions for each dataset. Further implementation details are in Appendix B.

## 3.2 BENCHMARK CONSTRUCTION

We introduce seven scenarios to evaluate MLLMs' proactiveness by drawing samples from diverse datasets, whose multi-choice options comprise proactive suggestions, the abstain option, and four categories, out of which only one is correct. Appendix B provides full details on each dataset.

**Moving occluding objects.** We repurposed the ROD (Lee et al., 2023) dataset by creating samples of 14 frames each, where the two possible suggestions are: moving the occluding object to the left or the right. The environment presents the model with a fully occluded image and a prompt, as Fig. 3 shows. Proactive suggestions tell the user to move occluding objects (e.g., the blue blocks) that obscure the object of interest (e.g., an orange), which the model aims to recognize. The model should ask to move the blocks, and, depending on the visibility of the occluded object, either predict its category or repeat the suggestion.

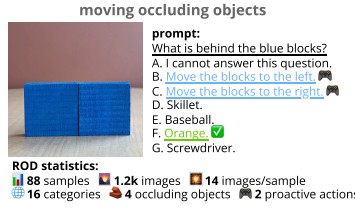

Figure 3: **ROD overview.**

**Handling temporal occlusions.** We repurposed VSOD (Liao et al., 2020), a dataset of public event videos with bounding boxes annotations for occlusions, to evaluate proactiveness under temporally evolving occlusions. We manually annotated public figures, the number of people in the scene, and the type of event for each frame, using them as the target category. Each sample contains, on average, about 230 image frames. As Fig. 4 shows, the environment initially shows the model the most occluded frame of the sample alongside valid options. The model should proactively suggest to the user to rewind the video or wait for the occlusion to disappear before answering.

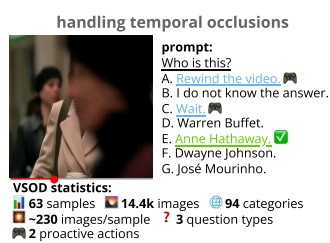

Figure 4: **VSOD overview.**

**Handling uninformative views.** We repurposed MVP-N (Wang et al., 2022a), a dataset containing multi-angle object views, to evaluate proactiveness in handling uninformative views. We built samples with one or more uninformative views followed by an informative one. As Fig. 5 shows, the environment returns the first image from a sample, which is not informative for predicting the correct target category. The model should proactively ask the user to rotate the object (or the camera) until it returns an informative view where the target category can be reliably predicted (e.g., Activia Yogurt Apple).

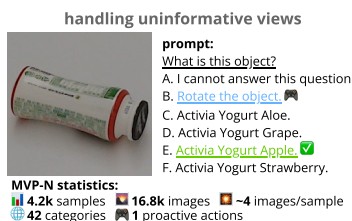

Figure 5: **MVP-N overview.**

**Improving image quality.** We repurposed ImageNet-C (IN-C) (Hendrycks & Dietterich, 2019) to test proactivess under corruptions, by creating samples where first and last images are the most and the least corrupted, respectively. As Fig. 6 shows, the environment returns a corrupted image (e.g., defocus blur), not suitable for predicting the correct category (e.g., White shark). The model should propose image quality enhancements (e.g., deblurring, reducing brightness, removing artifacts, increasing contrast) from a total of eight alternatives to improve the image quality. In the example of Fig. 6, the model should propose to deblur the image to predict the correct category.

**Asking for visual details.** Different from the previous cases, here we assess the model's proactiveness by its ability to propose proactive suggestions from a partial sketch. We repurposed the QuickDraw (QD) (Jongejan et al., 2016) dataset by rendering multiple PNGs for the same sketch, where each image includes one additional stroke compared to the previous one. The more strokes are added, the more recognizable the sketch becomes. As Fig. 7 shows, the environment first presents an image to the model that does not have enough detail to recognize the target category (e.g., clock). In this case, the proactive suggestion by the model is to improve the drawing, i.e., adding another stroke.

**Handling temporal ambiguities.** In this scenario, proactiveness is judged by the ability to seek information situated in a different instant of time in long videos. We repurposed the ChangeIt (CIT) dataset (Souček et al., 2022), consisting of videos of people interacting with objects, by creating samples of frames showing the objects' transformation (e.g., preparing tacos) from start to end. As Fig. 8 shows, the environment presents an input frame where the target category (e.g., tacos) is not visible. Similar to handling temporal occlusions, the proactive suggestion of the model is to ask the user to either rewind the video or wait for the informative moment to appear.

**Proposing camera movements.** Finally, we consider a practical scenario that prompts the user to spatially move the camera in a 3D plane to obtain more informative visual cues. We repurposed MS-COCO (Lin et al., 2014) to create samples containing different crops of the same image, where some crops are more informative than others. As Fig. 9 shows, the environment presents an uninformative crop to the model, where the target category (e.g., clock) is barely visible. The model should ask the user to move the camera or zoom to reveal the target object and answer the question. In the case on the right, the model should ask the user to move the camera towards the right.

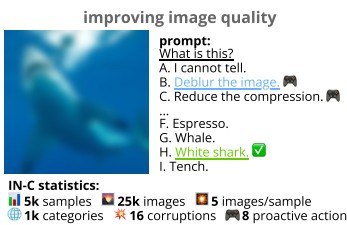

**improving image quality**

**prompt:**
What is this?
A. I cannot tell.
B. Deblur the image. 🎮
C. Reduce the compression. 🎮
...
F. Espresso.
G. Whale.
H. White shark. ✅
I. Tench.

**IN-C statistics:**
📊 **5k** samples  🖼️ **25k** images  🖼️ **5** images/sample
🌐 **1k** categories  ☀️ **16** corruptions  🎮 **8** proactive actions

Figure 6: **IN-C overview.**

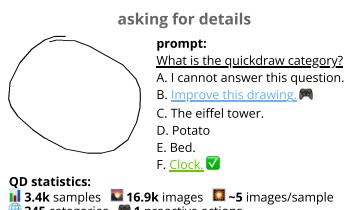

**asking for details**

**prompt:**
What is the quickdraw category?
A. I cannot answer this question.
B. Improve this drawing. 🎮
C. The eiffel tower.
D. Potato
E. Bed.
F. Clock. ✅

**QD statistics:**
📊 **3.4k** samples  🖼️ **16.9k** images  🖼️ **~5** images/sample
🌐 **345** categories  🎮 **1** proactive actions

Figure 7: **QD overview.**

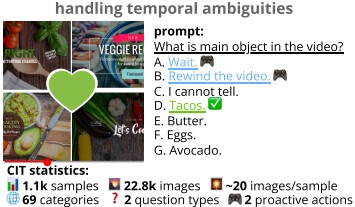

**handling temporal ambiguities**

**prompt:**
What is main object in the video?
A. Wait. 🎮
B. Rewind the video. 🎮
C. I cannot tell.
D. Tacos. ✅
E. Butter.
F. Eggs.
G. Avocado.

**CIT statistics:**
📊 **1.1k** samples  🖼️ **22.8k** images  🖼️ **~20** images/sample
🌐 **69** categories  ❓ **2** question types  🎮 **2** proactive actions

Figure 8: **CIT overview.**

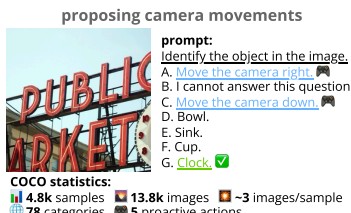

**proposing camera movements**

**prompt:**
Identify the object in the image.
A. Move the camera right. 🎮
B. I cannot answer this question.
C. Move the camera down. 🎮
D. Bowl.
E. Sink.
F. Cup.
G. Clock. ✅

**COCO statistics:**
📊 **4.8k** samples  🖼️ **13.8k** images  🖼️ **~3** images/sample
🌐 **78** categories  🎮 **5** proactive actions

Figure 9: **MS-COCO overview.**

### 3.3 FILTERING

As most of the datasets do not report how informative a frame is (except ROD and MVP-N), some images (e.g., 55.3% in ImageNet-C) can be correctly classified from the first frame. This allows models to bypass the need for human intervention, leading to uneven performance across tasks. To focus the evaluation on proactive behaviors, we filter out samples where the majority of MLLMs are capable of correctly guessing at the first turn, while *preserving* those that require multiple turns to be correctly classified. Therefore, we filter out samples that were correctly predicted at least 25% of the time *during the first turn*. After applying this filter, the average task accuracy in the first turn drops from 32.5% to 6.4%, thus requiring proactive suggestions to achieve good scores. The resulting filtered benchmark counts 7,557 samples from the original size of 17,909. For completeness, we discuss the filtering effect and results on unfiltered data in Sec. B.8.

Table 1: **MLLMs results on ProactiveBench.** We report the accuracy (*acc*) in percentages (%) and average number of proactive suggestions (*ps*) for all datasets, with global averages in the last column.

| family | model | ROD acc | ps | VSOD acc | ps | MVP-N acc | ps | IN-C acc | ps | QD acc | ps | CIT acc | ps | COCO acc | ps | avg. acc | ps |
|---|---|---|---|---|---|---|---|---|---|---|---|---|---|---|---|---|---|
| LLaVA-1.5 | 7B | 12.5 | 0.7 | 26.2 | 1.7 | 6.7 | 0.0 | 26.2 | 0.8 | 25.5 | 0.7 | **44.2** | 1.3 | 32.3 | 0.9 | 24.8 | 0.9 |
| LLaVA-NeXT | Mistral-7B | 0.0 | 0.0 | 0.0 | 0.2 | 1.6 | 0.1 | 10.2 | 0.4 | 1.0 | 0.1 | 17.2 | 1.4 | 1.6 | 0.0 | 4.5 | 0.3 |
|  | Vicuna-7B | 19.3 | 0.7 | 11.9 | 0.5 | 6.5 | 0.1 | 33.2 | 1.3 | 10.2 | 0.9 | 36.6 | 0.9 | 17.1 | 0.3 | 19.3 | 0.7 |
| LLaVA-OV | 0.5B | 44.3 | 2.3 | 9.5 | 1.6 | 12.8 | 0.4 | 24.8 | 1.4 | **33.8** | 1.5 | 31.1 | 1.4 | 16.9 | 0.4 | 24.8 | 1.3 |
|  | 7B | 0.0 | 0.0 | 14.3 | 0.4 | 6.7 | 0.0 | 27.8 | 1.0 | 24.3 | 0.4 | 10.4 | 0.3 | 3.2 | 0.0 | 12.4 | 0.3 |
|  | 72B | 0.0 | 0.0 | 19.0 | 0.4 | 5.0 | 0.1 | 32.2 | 1.2 | 14.3 | 0.2 | 16.9 | 0.5 | 3.7 | 0.0 | 13.0 | 0.3 |
| SmolVLM2 | 2.2B | 0.0 | 0.0 | 11.9 | 0.2 | 11.1 | 0.1 | 19.5 | 1.0 | 9.9 | 0.6 | 25.5 | 0.6 | 5.8 | 0.0 | 12.0 | 0.4 |
| Idefics3 | 8B | 31.8 | 1.6 | 19.0 | 2.2 | 7.4 | 0.1 | 32.1 | 1.1 | 12.5 | 0.6 | 12.1 | 0.4 | 9.0 | 0.0 | 17.7 | 0.9 |
| InstructBLIP | 7B | 0.0 | 0.0 | 9.5 | 1.3 | 8.8 | 0.1 | 11.3 | 0.0 | 18.3 | 0.1 | 24.5 | 0.0 | 12.6 | 0.0 | 12.2 | 0.2 |
| Qwen-2.5-VL | 3B | 0.0 | 0.0 | 9.5 | 0.0 | 4.9 | 0.0 | 35.9 | 2.0 | 7.9 | 0.2 | 12.4 | 0.3 | 6.3 | 0.0 | 11.0 | 0.4 |
|  | 7B | 0.0 | 0.0 | 0.0 | 0.0 | 4.3 | 0.0 | 40.5 | 1.3 | 9.9 | 0.1 | 9.8 | 0.1 | 4.9 | 0.0 | 9.9 | 0.2 |
|  | 32B | 0.0 | 0.0 | 4.8 | 0.0 | 4.6 | 0.0 | 30.9 | 0.4 | 12.3 | 0.0 | 17.4 | 0.4 | 5.5 | 0.0 | 10.8 | 0.1 |
|  | 72B | 0.0 | 0.0 | 2.4 | 0.2 | 6.7 | 0.0 | 29.2 | 0.9 | 3.1 | 0.1 | 9.3 | 0.3 | 2.0 | 0.0 | 7.5 | 0.2 |
| InternVL3 | 1B | **61.4** | 2.1 | 21.4 | 0.3 | 19.7 | 0.4 | 38.6 | 1.1 | 15.0 | 0.5 | 16.9 | 0.3 | 16.5 | 0.1 | 27.1 | 0.7 |
|  | 2B | 1.1 | 0.0 | **31.0** | 0.3 | **20.1** | 0.2 | 46.1 | 1.5 | 18.1 | 0.5 | 28.5 | 0.6 | 29.7 | 0.2 | 24.9 | 0.5 |
|  | 8B | 0.0 | 0.0 | 11.9 | 0.2 | 6.4 | 0.0 | 37.7 | 1.0 | 15.4 | 0.5 | 10.1 | 0.2 | 7.1 | 0.0 | 12.7 | 0.3 |
|  | 38B | 0.0 | 0.0 | **31.0** | 2.3 | 12.5 | 0.2 | 45.5 | 0.7 | 16.8 | 0.5 | 27.0 | 1.0 | 28.4 | 0.2 | 23.0 | 0.7 |
|  | 78B | 0.0 | 0.0 | 16.7 | 0.3 | 10.7 | 0.0 | 39.8 | 0.1 | 5.3 | 0.0 | 17.4 | 0.4 | 19.2 | 0.0 | 15.6 | 0.1 |
| Phi-4-Multimodal | 6B | 1.1 | 0.0 | 16.7 | 1.0 | 18.9 | 0.0 | 29.8 | 1.6 | 21.9 | 0.4 | 32.6 | 0.6 | 15.2 | 0.2 | 19.4 | 0.5 |
| OpenAI | GPT-4.1 | 0.0 | 0.0 | 0.0 | 0.2 | 15.2 | 0.1 | **68.2** | 1.1 | 15.0 | 0.2 | 23.5 | 0.6 | **94.4** | 0.0 | 30.9 | 0.3 |
|  | o4-mini | 0.0 | 0.0 | 16.7 | 0.6 | 19.8 | 0.0 | 49.0 | 0.2 | 21.6 | 0.0 | 37.9 | 0.8 | 92.8 | 0.0 | **34.0** | 0.2 |

## 4 EXPERIMENTS

Section 4.1 describes our evaluation protocol, tested models, and metrics used. Then, Sec. 4.2 describes ProactiveBench results, evaluating the proactiveness of several MLLMs. Finally, Sec. 4.3 reports additional ProactiveBench analysis, evaluating ways to elicit proactive suggestions.

### 4.1 EXPERIMENTAL SETUP

**Evaluation protocol.** For each evaluation step, we feed the MLLM with the user prompt (the question), the current image, and the valid set of suggestions, as Sec. 3.1 describes. Therefore, the multi-choice question prompt consists of three parts: the question, optionally a hint to elicit proactiveness, and the options (Sec. 3.2). Hints are dataset-specific and lead the model towards considering proactive suggestions (e.g., "Hint: rotating the object could provide a more informative view" for MVP-N). The conversation history is always discarded from one step to another unless explicitly mentioned (see Sec. 4.3). Finally, as VSOD and ChangeIt consist of video frames, we also tell the model that the visual input is taken from a video.

**Tested models.** We selected MLLMs from open and closed-weight models. Among open-weights models we chose recent and well-established ones: LLaVA-1.5 7B (Liu et al., 2024a), LLaVA-NeXT 7B (Liu et al., 2024a) with Mistral (Jiang, 2024) and Vicuna (Chiang et al., 2023) LLMs, LLaVA-OV 0.5B-7B-72B (Li et al., 2025), SmolVLM2 2.2B (Marafioti et al., 2025), Idefics3 8B (Laurençon et al., 2024), InstructBLIP (Dai et al., 2023), Qwen2.5-VL 3B-7B-32B-72B (Bai et al., 2025), InternVL3 1B-2B-8B-38B-78B (Zhu et al., 2025), Phi-4-Multimodal (Abouelenin et al., 2025). Finally, we picked GPT-4.1 and o4-mini (OpenAI, 2025b) among closed-weight models.

**Metrics.** For each model, we report the accuracy (*acc*), i.e., the percentage of correctly classified samples in a dataset over multiple turns, and the number of proactive suggestions (*ps*), namely the average number of human interventions requested by the model on a dataset. We also report the averaged metrics over the seven scenarios presented in Sec. 3.2.

### 4.2 MLLMs RESULTS IN PROACTIVEBENCH

Figure 10 compares models' accuracy (*acc*) obtained via Sec. 3 protocol and filtering, with the oracle setting, where we use a reference frame (i.e., with no occlusions or ambiguity). The goal of this comparison is to disentangle the recognition ability of MLLMs and their proactiveness. Results correspond to the average performance of all evaluated MLLMs. We notice a large discrepancy between the two settings. While MLLMs correctly classify 79.8% of samples in the oracle setting, they underperform by more than 60% when tasked with navigating to the correct answer through

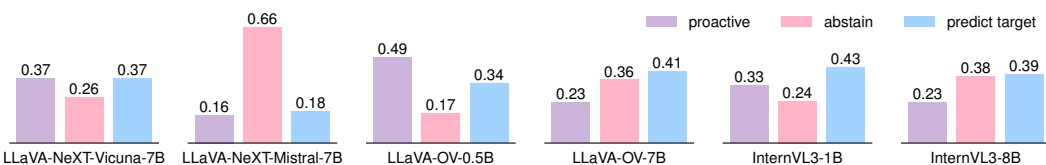

Figure 11: **Action distributions.** While LLaVA-OV 7B, InternVL3 8B, and LLaVA-NeXT Mistral tend to abstain or try to predict the correct answer, the other three models prefer to predict proactive suggestions over abstention, which can lead them to better visual cues and predict the correct action.

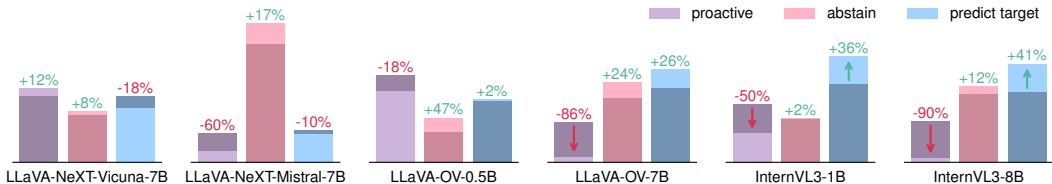

Figure 12: **Action distributions with random proactive options.** Lighter bars describe variations when using random, distracting, proactive suggestions.

proactive suggestions. The discrepancy is quite stark in the ROD dataset, where models achieve 8.2% accuracy, while the oracle counterpart reaches 98.3% on average. This demonstrates a severe lack of MLLMs' proactiveness. Models' individual performance in the oracle setting is in Appendix C.

Table 1 reports models' individual performance on ProactiveBench. Surprisingly, there is no clear correlation between model sizes and performance, e.g., InternVL3 1B outperforms InternVL3 8B in terms of accuracy (27.1% vs. 12.7%) and proactive suggestions (0.7 vs. 0.3). Furthermore, older models (e.g., LLaVA-1.5 7B) even outperform their newer and larger counterparts (i.e., LLaVA-OV 72B) by a discrete margin (24.8% vs. 13.0%) also in terms of *ps* (0.9 vs. 0.3). Interestingly, the LLM has an impact on the results, with LLaVA-NeXT Mistral achieving lower performance than its counterpart using Vicuna (4.5% vs. 19.3%). Instead, closed-source models (i.e., GPT-4.1 and o4-mini) show the best performance, with a low *ps* rate. Yet, they achieve extremely high accuracies on MS-COCO

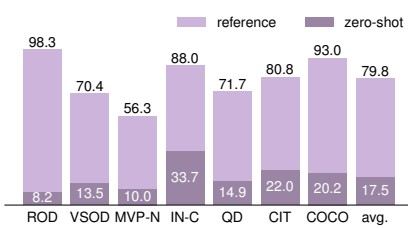

Figure 10: **Results vs. oracle performance (*acc*).** Models underperform by over 60% with ambiguous inputs.

(about $3\times$ better than other models), suggesting potential memorization caused by training data contamination. Unfortunately, we cannot verify this due to the proprietary nature of the data.

We investigate the unexpected behaviors above by visualizing the action distribution for proactive, abstain, and target category predictions in Fig. 11. Specifically, we compare pairs of MLLMs having different LLMs (i.e., LLaVA-NeXT Mistral and Vicuna) and different parameter counts (i.e., LLaVA-OV 0.5B and 7B, InternVL3 1B and 8B). While LLaVA-OV 7B, InternVL3 8B, and LLaVA-NeXT Mistral tend to abstain over sampling proactive suggestions, the other three show the exact opposite behavior. Thus, they are more likely to be proactive (over twice as likely for LLaVA-OV 0.5B) and, as a result, reach better states, leading to higher accuracy. A similar behavior was reported in (Wolfe et al., 2024), with LLaVA-NeXT Mistral abstaining more than LLaVA-NeXT Vicuna. Results for all models are reported in Appendix C.

### 4.3 ANALYZING AND ELICITING MLLMs PROACTIVENESS

**Are some MLLMs more proactive than others?** To answer this question, we replaced valid proactive suggestions with invalid ones chosen randomly from other datasets (e.g., "rewind the video" for QuickDraw). If a model chooses an invalid proactive option, this suggests that the model is not proactive but prefers to answer (even incorrectly) over abstaining. Figure 12 shows the action

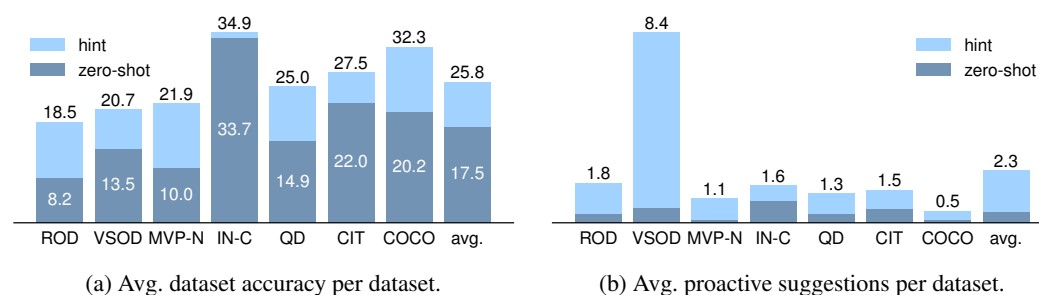

(a) Avg. dataset accuracy per dataset.      (b) Avg. proactive suggestions per dataset.

Figure 13: Performance when **conditioning action sampling with hints.** Results are averaged across all MLLMs. Zero-shot refers to models not prompted with hints.

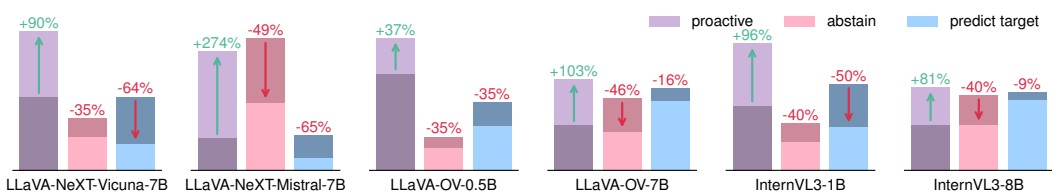

Figure 14: **Action distributions with hints.** Bars describe action distributions with (light) or without (dark) hints in the prompt. Hinting tilts the action distributions in favor of the proactive suggestion.

distribution for this experiment with the same six models as Fig. 11. Replacing valid proactive suggestions with invalid ones substantially reduces proactiveness for LLaVA-NeXT Mistral, LLaVA-OV 7B, and InternVL3 8B (i.e., -60%, -86%, and -90% relative decrease, respectively). Instead, other models seem less bothered by random practice options. LLaVA-NeXT Vicuna even increases the probability of sampling proactive suggestions (from 37% to 49%). These insights indicate that models showing a higher rate of proactive suggestions are **not** necessarily proactive, but rather they are less prone to abstain (Shukor et al., 2024), preferring unknown answers. Full results in Appendix C.

**Does hinting boost proactiveness?** Explicitly hinting at proactive suggestions may help navigate to the correct answer by eliciting MLLMs' proactiveness. To evaluate this hypothesis, we add **dataset-specific** hints to the prompt (e.g., "Hint: moving the occluding object might reveal what is behind it" for ROD), measuring how it affects the accuracy and number of proactive suggestions. We report the hints used in Appendix B. Figure 13b shows that hinting increases the proactive suggestions rate by 1.9 on average, with a significant boost in VSOD, likely caused by its numerous frames. Nonetheless, the accuracy does not surpass the random choice on average, reaching 25.8% (+8.3%). We also noticed that 16.0% of the time, MLLMs blindly chose proactive suggestions, disregarding the original task and reaching the maximum exploration steps allowed by the environment, thus failing to predict the correct category. Although hinting increases proactiveness, models may over-exploit proactive suggestions, failing to classify the object even if they stumble across the reference image. Figure 14 further visualizes this by showing how action distributions change using the same six models as Fig. 11. While original distributions (in darker colors) suggest that models infrequently choose proactive options, adding hints completely changes this behavior, preferring hinted actions over predicting the correct category. We report individual MLLMs performance in Appendix C.

**Does knowledge of the past elicit proactiveness?** Section 3.1 formalizes ProactiveBench evaluation, allowing MLLMs to observe the current state only. A key question is whether incorporating previous states and actions into the policy, i.e., $\pi_\theta(a_t|q, s_0, a_0, ..., s_t)$, elicits proactiveness. Thus, we keep the MLLM conversation history, limiting this evaluation to models supporting multi-image inference. Figure 15 shows the outcome of this experiment. The average accuracy drops by 7% while the number of proactive suggestions increases from 0.5 to 1.8 on average, compared to the zero-shot case. ROD average accuracy, in particular, is lowered by almost ten times (1.5% vs. 14.0%). Although models are not explicitly "told" to be proactive, like in Fig. 13, past proactive suggestions bias models towards repeating them. In fact, 12.9% of the time models displayed the same behavior as with "hints", where they repeatedly selected the proactive suggestions until reaching the maximum number

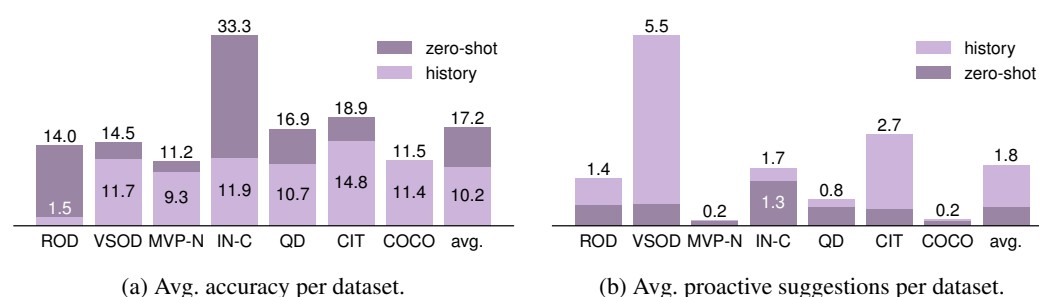

(a) Avg. accuracy per dataset.

(b) Avg. proactive suggestions per dataset.

Figure 15: Performance when **conditioning on conversation histories.** Results are averaged across all MLLMs. Zero-shot refers to models not integrating information about previous states.

of allowed steps. This value is lower compared to 16.0% of the previous case, as the first action is always unconditioned: thus, the blind selection of proactive actions only occurs if the first action is also proactive. We report models's individual performance in Appendix C.

**Do few-shot samples improve proactiveness?** We now investigate whether conditioning the policy on a few correct examples elicits proactiveness, improving accuracy. Let $c = (q^c, s_0^c, a_0^c, ..., s_t^c, a_c^c)$ be a conversation example leading to the correct answer $a_c^c$. We condition the action sampling on $m$ of such examples, $\pi_\theta(a_t|c_0, ..., c_m, q, s_t)$ on ROD and MVP-N, the only datasets supporting automatic few-shot sample generation (as image informativeness is annotated). We experiment with $m = 1$ and $m = 3$. Figure 16 shows how proactiveness changes with few-shot in-context learning (ICL).

Compared to the previous setting (indicated as zero-shot in the figure), the avg. *ps* increases by 1.4 and 0.2 on ROD and MVP-N, and 1.6 and 0.5 with one and three samples, respectively. Furthermore, the accuracy drops in ROD while remaining stable in MVP-N, resulting in 6.7% and 11.6% with one sample and 12.0% and 12.2% with three. When conditioning ROD experiments with one sample, we notice that models either tend to predict the same category of the ICL example or blindly select proactive suggestions until reaching the maximum number of exploration steps. Scaling ICL to three

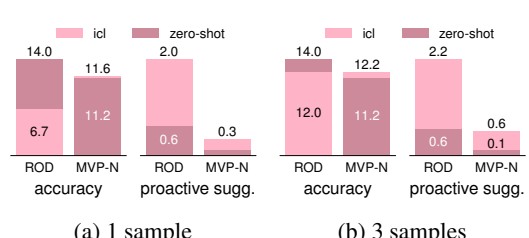

(a) 1 sample     (b) 3 samples

Figure 16: Performance when **conditioning on few shots.** Results are averaged across all MLLMs.

samples helps some models (e.g., LLaVA-OV 7B and Phi-4-Multimodal) predict the correct answer. Generally, InternVL3 1B and LLaVA-OV 0.5B are the most prone to consistently repeat proactive suggestions and disregard the main task, while InternVL3 8B and SmolVLM2 2.2B tend to abstain. Similarly, in MVP-N, model errors arise either from random guesses, abstentions, or, occasionally, valid proactive sequences ending with incorrect predictions. Full results are shown in Appendix C.

## 5   CONCLUSION

This paper presents ProactiveBench, a novel benchmark that evaluates MLLMs' proactiveness by pairing multi-choice questions with visual inputs that require human intervention (e.g., move the occluding object) to make the query answerable. We built ProactiveBench by repurposing seven existing datasets designed for different tasks, creating sequences that allow evaluating proactiveness in seven distinct scenarios in a multi-turn fashion. Our findings suggest that existing MLLMs are not proactive and prefer to abstain or predict random categories. Additionally, our analysis shows that hinting at the proactive action improves proactivity, with marginal accuracy gains. Furthermore, conditioning models on conversation histories and few-shot examples negatively biases the action distribution, with lower accuracy scores. These findings highlight ProactiveBench challenges, which we publicly release for future research.

## REPRODUCIBILITY STATEMENT

To ensure reproducibility of our results, we report the used code and the presented benchmark at: `https://anonymous.4open.science/r/ProactiveBench`

## ETHICS STATEMENT

All new material presented in this work is derived from existing, publicly available datasets.

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

# APPENDIX

This Appendix reports experiments, results, and implementation details that could not fit in the main paper. Appendix A evaluates MLLMs on ProactiveBench via free-form generation, further validating the findings of the main paper. Appendix B describes each dataset environment (Secs. B.1 to B.7), expanding main paper details, and provides further insights about the benchmark filtering (Sec. B.8). Instead, Appendix C supplements the main results with detailed tables of individual model performance. Finally, Appendices D to G report ProactiveBench broader impact statement, the licenses for the used datasets, models and ProactiveBench, additional qualitative examples, and the declaration about LLM usage.

## A  EVALUATION ON FREE-FORM GENERATION

This section evaluates multimodal LLMs on ProactiveBench via open-ended generation, validating the multiple-choice question-answering framework used in the main paper. Therefore, we only provide the MLLM with the image frame and the question that the model should answer.

**Evaluation protocol.** As evaluating free-form answers is challenging, we follow previous works (Liu et al., 2023a; Fu et al., 2024; Ma et al., 2024; Maaz et al., 2023; Song et al., 2024; Nagrani et al., 2024; Plizzari et al., 2025) and employ LLM-as-a-judge to provide a score to each answer. In particular, we use OpenAI's o4-mini (OpenAI, 2025a) and prompt it to spot proactive suggestions and category predictions. The following system and user prompts were used to query the judge:

```
System Prompt:
You are an automatic evaluation system.
You will receive:
- A user prompt (describing the task or question)
- A list of correct answers (accepted correct outputs)
- A system output (the model's generated answer)
Your task is to evaluate whether the system output includes at
least one of the correct answers clearly and correctly.

Guidelines:
- Minor wording differences (e.g., paraphrasing) are acceptable as
long as the meaning is preserved.
- If a correct answer is present but accompanied by incorrect or
irrelevant content, still consider it correct (score = 1).
- If none of the correct answers are present, or the output is
incorrect, mark it as wrong (score = 0).

Respond with a comma-separated list of 0s and 1s, one for each
correct answer in the list.\n\n
-------------------------------------------------------------------
User Prompt:
Input:
User Prompt:  {user_prompt}
Correct Answers:  {correct_answer}
System Output:  {generated_answer}

Your Response:  (Only output comma-separated list of 0s and 1s)
```

Thus, as answers are usually long, the LLM-as-a-judge is tasked to spot whether the answer contains correct proactive suggestions and correct category predictions, respectively defined in `{correct_answer}`. The judge returns comma-separated values, with one digit for each correct answer (either a valid proactive suggestion or the right category). Evaluating multi-turn conversations with free-form answers, however, is highly impractical since it requires alternating the answer generation with the MLLM and the evaluation with LLM-as-a-judge to update the environment state. Therefore, we limited this evaluation to single-turn conversations. Furthermore, to reduce the evaluation cost, we subsample each dataset to 500 entries, except ROD and VSOD, as they have smaller dimensions. We did not apply our filtering strategy as we noticed models could not easily

answer in the first turn. We report the ratio of correctly predicted categories ($acc$), the ratio of correct proactive suggestions ($pa$), and the aggregate accuracy ($agg$), computed as the average accuracy in either predicting the correct answer or providing a valid proactive suggestion.

**Results.** Table 2 shows the models' performance in open-ended generation on ProactiveBench. Due to the computational overhead and the monetary cost, we limit this evaluation to six representative models, i.e., LLaVA-OV 0.5B-7B, Qwen2.5-VL 3B-7B, and InternVL3 1B-8B. Overall, larger models tend to outperform smaller ones, e.g., LLaVA-OV 7B outperforms LLaVA-OV 0.5B, with the former being slightly more proactive than the smaller ones, except for InternVL3 1B proposing few correct proactive suggestions (i.e., 0.1% on avg.). By hinting at proactive suggestions (i.e., `If you cannot answer this question, please tell me what I should do to help you.`), the proactiveness increases for all MLLMs, with larger models showing higher absolute growths (e.g., Qwen2.5-VL 7B +10.5% vs. Qwen2.5-VL 3B +3.2%). Yet, performances are still very low, and the proactive suggestion rate grows only if models are explicitly told to be proactive, validating main paper's findings. Finally, we also notice that by hinting, models become more cautious in answering questions, showing lower ratios of correct predictions (e.g., from 15.2% to 12.7% of Qwen2.5-VL 7B).

Table 2: **Open-ended generation evaluation on ProactiveBench.** We report the aggregate accuracy ($agg$), the ratio of correctly predicted categories ($acc$), and the ratio of correct proactive suggestions ($pa$) for all datasets, with global averages in the last column.

| model | ROD | | | VSOD | | | MVP-N | | | IN-C | | |
|---|---|---|---|---|---|---|---|---|---|---|---|---|
| | agg | acc | pa | agg | acc | pa | agg | acc | pa | agg | acc | pa |
| LLaVA-OV-0.5B | 0.0 | **0.0** | 0.0 | 6.3 | 6.3 | **0.0** | 0.0 | **0.0** | 0.0 | 10.6 | 10.6 | 0.0 |
| LLaVA-OV-7B | **2.3** | 0.0 | **2.3** | 7.9 | 7.9 | 0.0 | 0.0 | 0.0 | 0.0 | 11.2 | 11.0 | **0.2** |
| Qwen2.5-VL-3B | 0.0 | 0.0 | 0.0 | **11.1** | **11.1** | 0.0 | 0.0 | 0.0 | 0.0 | 11.4 | 11.4 | 0.0 |
| Qwen2.5-VL-7B | 1.2 | 0.0 | 1.2 | 11.1 | 11.1 | 0.0 | 0.0 | 0.0 | 0.0 | **18.2** | **18.2** | 0.0 |
| InternVL3-1B | 0.0 | 0.0 | 0.0 | 7.9 | 7.9 | 0.0 | 0.0 | 0.0 | 0.0 | 12.4 | 12.4 | 0.0 |
| InternVL3-8B | 0.0 | 0.0 | 0.0 | 11.1 | 11.1 | 0.0 | 0.0 | 0.0 | 0.0 | 16.0 | 16.0 | 0.0 |

| model | QD | | | CIT | | | COCO | | | avg. | | |
|---|---|---|---|---|---|---|---|---|---|---|---|---|
| | agg | acc | pa | agg | acc | pa | agg | acc | pa | agg | acc | pa |
| LLaVA-OV-0.5B | 6.4 | 6.4 | 0.0 | 13.4 | 13.4 | 0.0 | 38.2 | 38.2 | **0.0** | 10.7 | 10.7 | 0.0 |
| LLaVA-OV-7B | **11.6** | **10.8** | 0.8 | 19.4 | 19.4 | 0.0 | 42.6 | 42.6 | 0.0 | 13.6 | 13.1 | **0.5** |
| Qwen2.5-VL-3B | 6.6 | 6.6 | 0.0 | 28.0 | 28.0 | 0.0 | 33.3 | 33.3 | 0.0 | 12.9 | 12.9 | 0.0 |
| Qwen2.5-VL-7B | 8.8 | 7.4 | **1.4** | 32.0 | 31.6 | **0.4** | 38.3 | 38.3 | 0.0 | **15.7** | 15.2 | 0.4 |
| InternVL3-1B | 3.4 | 3.0 | 0.4 | 29.4 | 29.4 | 0.0 | **45.6** | **45.6** | 0.0 | 14.1 | 14.0 | 0.1 |
| InternVL3-8B | 3.4 | 3.4 | 0.0 | **33.2** | **33.2** | 0.0 | 45.4 | 45.4 | 0.0 | 15.6 | **15.6** | 0.0 |

Table 3: **Open-ended generation evaluation on ProactiveBench by hinting at proactive suggestions.** We report the aggregate accuracy ($agg$), the ratio of correctly predicted categories ($acc$), and the ratio of correct proactive suggestions ($pa$) for all datasets, with global averages in the last column.

| model | ROD | | | VSOD | | | MVP-N | | | IN-C | | |
|---|---|---|---|---|---|---|---|---|---|---|---|---|
| | agg | acc | pa | agg | acc | pa | agg | acc | pa | agg | acc | pa |
| LLaVA-OV-0.5B | 1.1 | **0.0** | 1.1 | 6.3 | 6.3 | 0.0 | 0.0 | **0.0** | 0.0 | 8.2 | 8.2 | 0.0 |
| LLaVA-OV-7B | 0.0 | 0.0 | 0.0 | 4.8 | 4.8 | 0.0 | 0.4 | 0.0 | 0.4 | 28.2 | 8.4 | 20.0 |
| Qwen2.5-VL-3B | 5.7 | 0.0 | 5.7 | 6.3 | 6.3 | 0.0 | 0.2 | 0.0 | 0.2 | 17.8 | 9.4 | 8.4 |
| Qwen2.5-VL-7B | 1.1 | 0.0 | 1.1 | 7.9 | 6.3 | **1.6** | **1.2** | 0.0 | **1.2** | **40.2** | **15.4** | 25.2 |
| InternVL3-1B | 0.0 | 0.0 | 0.0 | **11.1** | **11.1** | 0.0 | 0.0 | 0.0 | 0.0 | 12.0 | 9.8 | 2.2 |
| InternVL3-8B | **27.3** | 0.0 | **27.3** | 6.3 | 6.3 | 0.0 | 0.6 | 0.0 | 0.6 | 30.4 | 13.0 | 18.4 |

| model | QD | | | CIT | | | COCO | | | avg. | | |
|---|---|---|---|---|---|---|---|---|---|---|---|---|
| | agg | acc | pa | agg | acc | pa | agg | acc | pa | agg | acc | pa |
| LLaVA-OV-0.5B | 7.6 | 7.4 | 0.2 | 12.4 | 12.4 | 0.0 | 39.2 | 39.2 | 0.0 | 10.7 | 10.5 | 0.2 |
| LLaVA-OV-7B | 23.5 | **8.9** | 14.8 | 18.6 | 18.6 | 0.0 | 31.8 | 30.6 | 1.2 | 15.3 | 10.2 | 5.2 |
| Qwen2.5-VL-3B | 14.2 | 6.6 | 8.2 | 24.4 | 24.2 | 0.2 | 28.4 | 28.4 | 0.0 | 13.9 | 10.7 | 3.2 |
| Qwen2.5-VL-7B | **42.5** | 6.8 | **39.1** | 33.4 | 26.8 | **6.8** | 34.4 | 33.3 | **1.4** | 23.0 | 12.7 | **10.9** |
| InternVL3-1B | 10.2 | 3.7 | 6.5 | 25.6 | 25.6 | 0.0 | 43.2 | 43.2 | 0.0 | 14.6 | 13.3 | 1.2 |
| InternVL3-8B | 25.8 | 3.2 | 23.3 | **34.4** | **32.0** | 2.6 | **45.0** | **44.8** | 0.2 | **24.3** | **14.2** | 10.3 |

# B  DATASET DETAILS AND ENVIRONMENT IMPLEMENTATION

This section expands Secs. 3.1 to 3.3, providing further information about data generation pipelines, environment details, and filtering.

## B.1 THE ROD ENVIRONMENT

The ROD (Lee et al., 2023) environment evaluates MLLMs' proactiveness in proposing to move occluding objects before answering the question. The first frame in the ROD environment depicts an occluding object that completely hides another object, as Fig. 22 shows. Each MLLM is prompted to predict the category of the occluded object, choosing out of four possible categories, and the abstain option. As the posed question is unanswerable from the initial frame, given that the subject of the question is invisible, the environment also returns two valid proactive suggestions among other options, i.e., `move the {occluding_object} to the left`, and `move the {occluding_object} to the right`, where `{occluding_object}` is replaced with the occluding object description (e.g., red cardboard, blue blocks). Furthermore, we also consider camera movement a valid proactive suggestion in the free-form evaluation experiments. A typical prompt is structured as follows:

```
Could you tell me what is behind the {occluding_object}?  <hint>
Choose from the following options.  Options:
A. Move the {occluding_object} to the left.
B. Move the {occluding_object} to the right.
C. {abstain option}.
D. {wrong random category}.
E. {wrong random category}.
F. {correct category}.
G. {wrong random category}.
Please only return one of the options without any other words.
```

The question is sampled from a pool of 15 similar questions generated by ChatGPT, and the abstain option is from a pool of three. Additionally, the first three options and the remaining four are shuffled, so the same option does not always appear in the same position. Shuffling is performed during data generation, resulting in a fixed order for each sample. Finally, `<hint>` indicates the position of the hint used in the main paper experiments (Sec. 4.3), which, in the case of ROD, corresponds to "`Hint: moving the occluding object might reveal what is behind it.`"

The set of valid actions $\mathcal{A}_t$ is constant throughout the evaluation, and MLLMs are allowed to move the occluding object 14 times, corresponding to the total number of frames for each sample. As the first frame is completely occluded, if a model predicts a category for the first frame, we count the prediction as wrong, as the first frame does not contain information about the target object class. After seven consecutive right or left movements from the most occluded frame, MLLMs encounter the reference frame, where the object is perfectly visible. Finally, the environment is circular, which means that by pursuing the same proactive suggestion, the occluding object will reveal the object until it reappears from the opposite side, gradually re-occluding the object.

## B.2 THE VSOD ENVIRONMENT

The VSOD environment evaluates MLLMs' proactiveness in proposing to wait or rewind the video before answering the question, in case of occlusions. The first frame in this environment depicts a scene where individuals are occluded by someone passing in front of the camera, as Fig. 23 shows. Each MLLM is prompted to predict the speaker's name, the number of people, or the event type, choosing out of four possible categories, and the abstain option. As the posed question is likely unanswerable from the initial frame, given that the subject of the question is (partially) invisible, the environment also returns two valid proactive suggestions among other options, i.e., `wait for the occlusion to disappear`, and `rewind the video`. Furthermore, we also consider camera movement a valid proactive suggestion in the free-form evaluation experiments. A typical prompt is structured as follows:

```
This is a frame extracted from a video.  Answer the following
question.  Could you tell me who is talking?  <hint> Choose from
the following options.  Options:
A. Rewind the video.
B. {abstain option}.
C. Wait for the occlusion to disappear.
D. {wrong random category}.
E. {correct category}.
F. {wrong random category}.
G. {wrong random category}.
Please only return one of the options without any other words.
```

In this prompt, the question is sampled from a pool of 45 similar questions (15 for each question type), and the abstain option is from a pool of three. Additionally, the first three options and the remaining four are shuffled, so the same option does not always appear in the same position. Shuffling is performed during data generation, resulting in a fixed order for each sample. Finally, `<hint>` indicates the position of the hint used in the main paper experiments (Sec. 4.3), which, in the case of VSOD, corresponds to "`Hint:  If there is an occlusion, waiting for it to disappear or rewinding the video might reveal what's behind it.`"

The set of valid actions $\mathcal{A}_t$ is constant throughout the evaluation, and MLLMs are allowed to propose proactive suggestions as many times as the number of frames in the video. As each occlusion lasts for a different amount of time, the number of proactive suggestions to reach a state where the question becomes answerable varies from sample to sample. Finally, if the MLLM suggests waiting at the last frame, we treat the sequence as circular and return the first frames. Analogously, we return the final frame if, at the first frame, the model suggests rewinding the video.

### B.3   THE MVP-N ENVIRONMENT

The MVP-N environment evaluates MLLMs' proactiveness in suggesting objects and camera rotations before answering the question in case of uninformative views. The first frame in the MVP-N environment depicts an object from an uninformative viewpoint, as Fig. 24 shows. Each MLLM is prompted to predict the category of the object, choosing out of four possible categories, and the abstain option. As the posed question is unanswerable from the initial frame, given that discriminative object features are invisible, the environment also returns a valid proactive suggestion among other options, e.g., `rotate the object`, `give me a view of the object from a different perspective`. As object orientation and camera extrinsic parameters are not annotated, the proactive suggestion is sampled from a pool of 11 prompts generated with ChatGPT that contain both object rotations and camera movements. A typical prompt is structured as follows:

```
Identify the object in this image.  <hint> Choose from the
following options.  Options:
A. {abstain option}.
B. {proactive suggestion}.
C. {wrong random category}.
D. {correct category}.
E. {wrong random category}.
F. {wrong random category}.
Please only return one of the options without any other words.
```

In this prompt, the question is sampled from a pool of 15 similar questions, and the abstain option is from a pool of three. Additionally, the first two options and the remaining four are shuffled, so the same option does not always appear in the same position. Shuffling is performed during data generation, resulting in a fixed order for each sample. Finally, `<hint>` indicates the position of the hint used in the main paper experiments (Sec. 4.3), which, in the case of MVP-N, corresponds to "`Hint: rotating the object could provide a more informative view.`"

The set of valid actions $\mathcal{A}_t$ is constant throughout the evaluation, and, since we generated sequences of various lengths, MLLMs are allowed to rotate the object or change camera angle 3 times on average for each sample, depending on the sequence. To find the informative view, MLLMs must

propose object rotations or camera movements until they reach the last state, where the object is distinguishable.

## B.4 THE IMAGENET-C ENVIRONMENT

The ImageNet-C environment evaluates MLLMs' proactiveness in suggesting image quality improvements before answering the question, in case of badly corrupted pictures. The first image in the ImageNet-C environment depicts one of ImageNet (Russakovsky et al., 2015) validation samples strongly corrupted by one of eight different corruptions, as Fig. 25 shows. Each MLLM is prompted to predict the category of the corrupted object, choosing out of four possible categories, and the abstain option. As the posed question is hardly answerable from the initial picture, the environment also returns four proactive suggestions, out of which only one is valid, e.g., `deblur the image`, `denoise the image`, `remove artifacts`. For example, a typical prompt is structured as follows:

```
What type of object do you see here?  <hint> Choose from the
following options.  Options:
A. {invalid proactive suggestion}.
B. {abstain option}.
C. {valid proactive suggestion}.
D. {invalid proactive suggestion}.
E. {invalid proactive suggestion}.
F. {wrong random category}.
G. {correct category}.
H. {wrong random category}.
I. {wrong random category}.
Please only return one of the options without any other words.
```

In this prompt, the question is sampled from a pool of 15 similar questions, and the abstain option is from a pool of three. Additionally, the first five options and the remaining four are shuffled, so the same option does not always appear in the same position. Shuffling is performed during data generation, resulting in a fixed order for each sample. Finally, `<hint>` indicates the position of the hint used in the main paper experiments (Sec. 4.3), which, in the case of ImageNet-C, corresponds to "`Hint: enhancing the image quality could help with classification.`"

As ImageNet-C counts 50,000 images, we subsampled 5 images per class, resulting in 5,000 images, making this dataset comparable in size to the others used. The set of valid actions $\mathcal{A}_t$ is constant throughout the evaluation, and MLLMs are allowed to propose the correct proactive suggestion 4 times, improving the image quality. After 4 proactive suggestions, MLLMs encounter the last frame, the reference one. Further proactive suggestions result in terminating the evaluation.

## B.5 THE QUICKDRAW ENVIRONMENT

The QuickDraw environment evaluates MLLMs' proactiveness in proposing to add details to a sketch, to make it more recognizable. The first image in the QuickDraw environment shows the first drawn stroke by a user in trying to depict a target object, as Fig. 26 shows. Each MLLM is prompted to predict the category of such depicted object, choosing out of four possible categories, and the abstain option. As the posed question is likely unanswerable from the initial drawing, the environment also returns a valid proactive suggestion among other options, e.g., `add more details`, or `could you improve the quickdraw?` For example, a typical prompt is structured as follows:

```
What is the category of the depicted object?  <hint> Choose from
the following options.  Options:
A. {proactive option}.
B. {abstain option}.
C. {wrong random category}.
D. {wrong random category}.
E. {wrong random category}.
F. {correct category}.
Please only return one of the options without any other words.
```

In this prompt, the question is sampled from a pool of 15 similar questions, the abstain option is from a pool of three, and the proactive option is from a pool of 13. Additionally, the first two options and the remaining four are shuffled, so the same option does not always appear in the same position. Shuffling is performed during data generation, resulting in a fixed order for each sample. Finally, `<hint>` indicates the position of the hint used in the main paper experiments (Sec. 4.3), which, in the case of QuickDraw, corresponds to "`Hint:  Adding more details to the quickdraw could help with classification.`"

As each drawing is also evaluated by a classification model (Jongejan et al., 2016), we discarded all drawings not recognized by such a model, avoiding unrecognizable drawings. Furthermore, the dataset contains 50 million drawings over 345 classes. Evaluating each MLLM would require approximately 300 GPU days. Thus, we subsample it to 10 samples per class, resulting in 3450 drawings. The set of valid actions $\mathcal{A}_t$ is constant throughout the evaluation, and MLLMs are allowed to ask for details a limited number of times, which depends on the number of strokes drawn by the user. Depending on the number of strokes, after requesting further details enough times, MLLMs encounter the reference frame, where the object is recognizable.

## B.6    THE CHANGEIT ENVIRONMENT

The ChangeIt environment evaluates MLLMs' proactiveness in proposing to seek the answer at a different moment in the video. The first frame in the ChangeIt environment shows the beginning of a video tutorial, as Fig. 27 shows. Each MLLM is prompted to either predict the category of the main object or the main action taken in the video, choosing out of four possible categories and the abstain option. As the posed question is likely unanswerable from the initial frame, the environment also returns two valid proactive suggestions among other options, i.e., `wait for the occlusion to disappear`, and `rewind the video`. For example, a typical prompt is structured as follows:

```
What action is being performed in the video?  <hint> Choose from
the following options.  Options:
A. Rewind the video.
B. Wait for the occlusion to disappear.
C. {abstain option}.
D. {wrong random category}.
E. {wrong random category}.
F. {wrong random category}.
G. {correct category}.
Please only return one of the options without any other words.
```

For this prompt, questions related to the object category are sampled from a pool of 15 similar questions, while those related to the action category are from a pool of 11 questions, all obtained by querying ChatGPT. The abstain option, instead, is sampled from a pool of three. Additionally, the first three options and the remaining four are shuffled, so the same option does not always appear in the same position. Shuffling is performed during data generation, resulting in a fixed order for each sample. Finally, `<hint>` indicates the position of the hint used in the main paper experiments (Sec. 4.3), which, in the case of ChangeIt, corresponds to "`Hint:  If you cannot answer the question, waiting for it to appear or rewinding the video could help with classification.`"

The set of valid actions $\mathcal{A}_t$ changes throughout the evaluation. Since the environment returns the initial frame first, the rewind option is disabled at the first frame and enabled from the second step. MLLMs can propose proactive suggestions as many times as the number of frames in the video. Finally, as each video differs, the number of proactive suggestions to reach a state where the question becomes answerable varies from sample to sample.

### B.7 THE MS-COCO ENVIRONMENT

The MS-COCO environment evaluates MLLMs' proactiveness in proposing camera movements to obtain more informative cues. The first image in the MS-COCO environment shows a trimmed picture with missing object details, as in Fig. 28. Since most images in MS-COCO contain multiple objects, we discard all those samples that contain more than one object, avoiding ambiguities. Each MLLM is prompted to predict the category of the object in the image, choosing out of four possible categories and the abstain option. As the posed question is likely unanswerable from the initial frame, the environment also returns one or two valid proactive suggestions, depending on how the image crop was computed. Crops are generated to allow for exploration of one of the ordinal or cardinal directions or zooming out, the set of proactive actions, thus, changes based on the picture, i.e., `move the camera up`, `move the camera down`, `move the camera left`, `move the camera right`, and `move farther from the object`. In the case of ordinal directions, MLLMs receive two proactive options, one for each of the cardinal directions that generate the ordinal one. Instead, for cardinal directions and zooming out, MLLMs receive only one. For example, a typical prompt for an ordinal direction is structured as follows:

```
Classify the visual content of this image.  <hint> Choose from the
following options.  Options:
A. Move the camera left.
B. Move the camera up.
C. {abstain option}.
D. {wrong random category}.
E. {wrong random category}.
F. {wrong random category}.
G. {correct category}.
Please only return one of the options without any other words.
```

For this prompt, the question is sampled from a pool of 15 similar questions obtained from querying ChatGPT, while the abstain option is sampled from a pool of three. Additionally, the first two/three options (depending on the direction) and the remaining four are shuffled, so the same option does not always appear in the same position. Shuffling is performed during data generation, resulting in a fixed order for each sample. Finally, `<hint>` indicates the position of the hint used in the main paper experiments (Sec. 4.3), which, in the case of MS-COCO, corresponds to "`Hint:  moving the camera could help with classification`" for ordinal and cardinal directions and "`Hint:  zooming out could help with classification`" for the zooming out case.

The set of valid actions $\mathcal{A}_t$ changes throughout the evaluation for ordinal directions, while it remains fixed for cardinal directions and the zooming out case. Since the camera can move in two of the four cardinal directions in the ordinal directions case, we remove a cardinal direction if the MLLM has already unveiled all possible object details in a specific direction, i.e., it has explored all discrete steps in a direction. Finally, MLLMs can propose proactive suggestions as many as the predefined discrete steps, set between 3 and 5.

### B.8 FILTERING

This section visualizes the effects of the filtering procedure (Sec. 3.3), showing for each dataset the number of remaining samples and the average accuracy at different filtering thresholds. Figure 17 shows for each dataset the original dataset size and the size after filtering. Datasets with images that are generally easier to classify correctly in the first turn undergo a larger reduction (e.g., IN-C decreases from 4,856 to 1,095 samples). Instead, Fig. 18 reports, for each dataset, the average accuracy at the first turn pre- and post-filtering, and compares them with the original and post-filtering

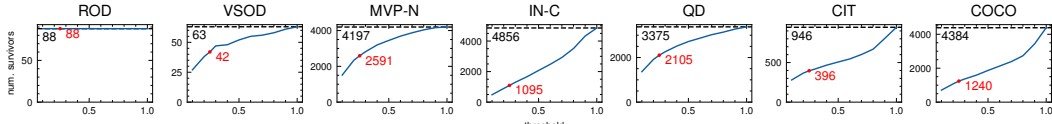

Figure 17: **Samples after filtering.** Each plot shows the remaining examples for each datasets after filtering. The light blue line represents the number of remaining examples at different thresholds. Instead, in black and red we report the original and post-filtering dataset size, respectively.

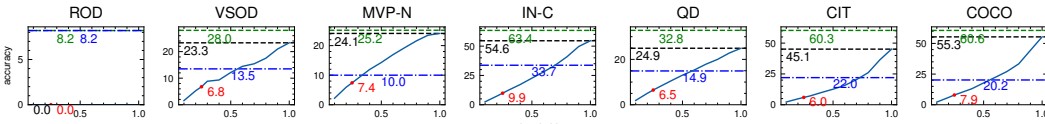

Figure 18: **Accuracy pre- and post-filtering.** Each plot shows the average MLLM's accuracy in the first turn for different datasets. The light blue line represents the accuracy at different thresholds, while in black and red we report the original and post-filtering accuracy, respectively. Finally, we report the multi-turn accuracy before and after filtering in green and blue.

zero-shot accuracy over multiple rounds. Finally, Tab. 4 reports MLLMs' zero-shot performance on the unfiltered benchmark.

## C EXTENDED RESULTS

As most results could not fit within nine pages, the main paper summarizes key findings with plots. This section reports all tables associated with the main paper's plots and the extended version of each plot, not limited to six models. Table 5 reports MLLMs oracle performance on ProactiveBench. Figure 19 shows the action distribution for all models, further highlighting that some overweight proactive suggestions over the abstain option. InternVL3 78B stands out, showing the lowest rate of proactive suggestions (4%), despite being one of the best open-weight MLLMs. Table 6 and Fig. 20 respectively report MLLM's results and the corresponding action distribution when proactive suggestions are replaced with random ones. Similarly, Tab. 7 and Fig. 21 describe MLLM's results and action distribution on all models when the prompt hints at proactive suggestions. Finally, Tab. 8 and Tab. 9 display MLLM's performance on ProactiveBench when conditioned on conversation histories and few-shot examples of proactive conversations. We do not report results for all models for experiments with random proactive options, conversation histories, and in-context learning, due to computational and monetary constraints.

**Computational details.** We conducted most experiments using a single A100 Nvidia GPU, 32GB of RAM, and 8 CPU cores, lasting about 1 hour, depending on the dataset. When conditioning on conversation histories and few-shot samples, we used two A100 GPUs to reduce the memory footprint of the models' parameters, with experiments lasting about 2 hours on average and at most 8 hours, depending on the dataset and model. Furthermore, to avoid out-of-memory issues for Phi-4-Multimodal with ICL examples, we reduced the ROD image sizes of the few shots from $3024 \times 3024$ to $512 \times 512$, and the sequence length of MVP-N to 2 when using 3 shots. Finally, we resized all samples' short edge to 224px when conditioning on conversational histories to avoid out-of-memory issues with long sequences.

## D BROADER IMPACTS STATEMENT

ProactiveBench is designed to assess the proactiveness of multimodal large language models (MLLMs), i.e., their ability to request additional input when faced with ambiguous or insufficient visual information. As MLLMs are increasingly deployed in interactive and safety-critical applications (i.e., assistive tools, autonomous systems), encouraging and evaluating such behavior is essential for developing more collaborative and user-aligned AI.

Table 4: **MLLMs results on unfiltered ProactiveBench.** We report the accuracy (*acc*) in percentages (%) and average number of proactive suggestions (*ps*) for all datasets without filtering, with global averages in the last column.

| family | model | ROD | | VSOD | | MVP-N | | IN-C | | QD | | CIT | | COCO | | avg. | |
|---|---|---|---|---|---|---|---|---|---|---|---|---|---|---|---|---|---|
| | | acc | ps | acc | ps | acc | ps | acc | ps | acc | ps | acc | ps | acc | ps | acc | ps |
| LLaVA-1.5 | 7B | 12.5 | 0.7 | 41.3 | 1.3 | 27.7 | 0.0 | 59.4 | 0.4 | 43.0 | 0.5 | 70.3 | 0.7 | 67.6 | 0.4 | 46.0 | 0.6 |
| LLaVA-NeXT | Mistral-7B | 0.0 | 0.0 | 9.5 | 0.2 | 13.7 | 0.1 | 53.9 | 0.2 | 12.2 | 0.1 | 46.3 | 1.4 | 49.1 | 0.0 | 26.4 | 0.3 |
| | Vicuna-7B | 19.3 | 0.7 | 25.4 | 0.9 | 26.2 | 0.1 | 69.2 | 0.5 | 22.0 | 0.7 | 68.6 | 0.4 | 67.7 | 0.1 | 42.6 | 0.5 |
| LLaVA-OV | 0.5B | 44.3 | 2.3 | 20.6 | 1.9 | 30.7 | 0.4 | 53.6 | 0.7 | 45.8 | 1.1 | 59.0 | 0.6 | 61.0 | 0.1 | 45.0 | 1.0 |
| | 7B | 0.0 | 0.0 | 30.2 | 0.3 | 24.2 | 0.0 | 70.3 | 0.4 | 46.7 | 0.3 | 56.4 | 0.1 | 60.0 | 0.0 | 41.1 | 0.0 |
| | 72B | 0.0 | 0.0 | 41.3 | 0.3 | 23.7 | 0.0 | 74.6 | 0.4 | 39.0 | 0.1 | 61.9 | 0.2 | 59.7 | 0.0 | 42.9 | 0.1 |
| SmolVLM2 | 2.2B | 0.0 | 0.0 | 23.8 | 0.3 | 26.6 | 0.0 | 55.8 | 0.5 | 27.0 | 0.5 | 64.0 | 0.3 | 59.9 | 0.0 | 36.7 | 0.2 |
| Idefics3 | 8B | 31.8 | 1.6 | 31.7 | 2.1 | 27.7 | 0.1 | 70.0 | 0.4 | 27.9 | 0.5 | 58.0 | 0.2 | 62.2 | 0.1 | 44.2 | 0.7 |
| InstructBLIP | 7B | 0.0 | 0.0 | 12.7 | 1.5 | 12.8 | 0.1 | 26.0 | 0.1 | 18.9 | 0.1 | 47.6 | 0.1 | 26.7 | 0.0 | 20.7 | 0.3 |
| Qwen-2.5-VL | 3B | 0.0 | 0.0 | 31.7 | 0.0 | 25.4 | 0.0 | 69.5 | 0.9 | 29.1 | 0.1 | 58.9 | 0.2 | 56.6 | 0.0 | 38.7 | 0.2 |
| | 7B | 0.0 | 0.0 | 17.5 | 0.0 | 24.7 | 0.0 | 78.5 | 0.5 | 34.3 | 0.1 | 60.6 | 0.0 | 59.5 | 0.0 | 39.3 | 0.1 |
| | 32B | 0.0 | 0.0 | 20.6 | 0.0 | 24.9 | 0.0 | 73.6 | 0.1 | 36.4 | 0.0 | 64.1 | 0.2 | 58.1 | 0.0 | 39.7 | 0.0 |
| | 72B | 0.0 | 0.0 | 20.6 | 0.6 | 27.4 | 0.0 | 72.0 | 0.2 | 25.3 | 0.0 | 55.0 | 0.1 | 55.1 | 0.0 | 36.5 | 0.1 |
| InternVL3 | 1B | **61.4** | 2.1 | 39.7 | 0.2 | 29.3 | 0.3 | 69.4 | 0.5 | 29.2 | 0.4 | 61.3 | 0.1 | 69.6 | 0.0 | **51.4** | 0.5 |
| | 2B | 1.1 | 0.0 | **49.2** | 0.2 | 30.5 | 0.1 | 76.9 | 0.6 | 37.9 | 0.4 | 69.3 | 0.3 | 77.1 | 0.1 | 48.9 | 0.2 |
| | 8B | 0.0 | 0.0 | 31.7 | 0.1 | 23.2 | 0.0 | 75.9 | 0.3 | 36.0 | 0.3 | 58.3 | 0.1 | 67.1 | 0.0 | 41.7 | 0.1 |
| | 38B | 0.0 | 0.0 | 44.4 | 1.7 | 31.4 | 0.1 | 84.4 | 0.2 | 39.1 | 0.3 | 68.8 | 0.5 | 77.4 | 0.0 | 49.4 | 0.4 |
| | 78B | 0.0 | 0.0 | 39.7 | 0.3 | 31.7 | 0.0 | 83.4 | 0.0 | 29.5 | 0.0 | 62.9 | 0.2 | 74.9 | 0.0 | 46.0 | 0.1 |
| Phi-4-Multimodal | 6B | 1.1 | 0.0 | 27.0 | 0.7 | 29.5 | 0.0 | 66.4 | 0.7 | 42.3 | 0.3 | 66.0 | 0.2 | 64.6 | 0.1 | 42.4 | 0.3 |
| OpenAI | o4-mini | 0.0 | 0.0 | 23.8 | 0.4 | **34.6** | 0.0 | 80.2 | 0.1 | 42.4 | 0.0 | **71.8** | 0.4 | 96.6 | 0.0 | 49.9 | 0.1 |
| | GPT-4.1 | 0.0 | 0.0 | 9.5 | 0.1 | 24.8 | 0.1 | **90.0** | 0.3 | 35.0 | 0.1 | 62.0 | 0.3 | **96.8** | 0.0 | 45.4 | 0.1 |

Table 5: **MLLMs oracle performance on ProactiveBench.** We report the accuracy in percentages (%) for all datasets, with global averages in the column.

| family | model | ROD | VSOD | MVP-N | IN-C | QD | CIT | COCO | avg. |
|---|---|---|---|---|---|---|---|---|---|
| LLaVA-1.5 | 7B | **100.0** | 76.2 | 32.6 | 91.0 | 72.9 | 76.8 | 93.0 | 77.5 |
| LLaVA-NeXT | Mistral-7B | **100.0** | 57.1 | 43.6 | 88.6 | 65.6 | 75.3 | 95.5 | 75.1 |
| | Vicuna-7B | 98.9 | 57.1 | 36.6 | 90.6 | 56.8 | 74.2 | 95.2 | 72.8 |
| LLaVA-OV | 0.5B | **100.0** | 40.5 | 60.7 | 84.6 | 78.1 | 78.5 | 96.0 | 76.9 |
| | 7B | **100.0** | 78.6 | 63.2 | 94.5 | 86.4 | 87.4 | 97.6 | 86.8 |
| | 72B | **100.0** | 83.3 | 68.0 | 95.3 | 88.1 | 87.1 | 97.6 | 88.5 |
| SmolVLM2 | 2.2B | **100.0** | 69.0 | 50.4 | 88.9 | 73.1 | 84.6 | 95.8 | 80.3 |
| Idefics3 | 8B | **100.0** | 76.2 | 52.5 | 90.4 | 67.5 | 83.6 | 96.1 | 80.9 |
| InstructBLIP | 7B | 75.0 | 57.1 | 21.5 | 31.5 | 32.5 | 61.4 | 25.6 | 43.5 |
| Qwen-2.5-VL | 3B | **100.0** | 78.6 | 51.7 | 91.5 | 65.5 | 84.8 | 96.0 | 81.2 |
| | 7B | **100.0** | 81.0 | 63.3 | 95.0 | 75.3 | **87.9** | 97.1 | 85.6 |
| | 32B | **100.0** | 78.6 | 53.8 | 93.2 | 72.3 | 84.3 | 95.8 | 82.6 |
| | 72B | **100.0** | 76.2 | 63.8 | 94.7 | 80.3 | 84.8 | 97.3 | 85.3 |
| InternVL3 | 1B | 98.9 | 52.4 | 55.4 | 88.0 | 62.5 | 80.3 | 96.5 | 76.3 |
| | 2B | **100.0** | 76.2 | 57.0 | 92.7 | 65.8 | 84.1 | 97.6 | 81.9 |
| | 8B | **100.0** | 76.2 | 59.9 | 96.1 | 70.1 | 84.1 | 97.6 | 83.4 |
| | 38B | **100.0** | 81.0 | 72.2 | 97.5 | 80.7 | 86.1 | 97.8 | 87.9 |
| | 78B | **100.0** | **85.7** | 74.5 | **98.3** | 80.9 | 87.4 | **98.7** | **89.4** |
| Phi-4-Multimodal | 6B | **100.0** | 57.1 | 47.5 | 82.6 | 77.9 | 74.2 | 96.0 | 76.5 |
| OpenAI | GPT-4.1 | **100.0** | 76.2 | **80.8** | 98.2 | **88.2** | 83.6 | 96.8 | 89.1 |
| | o4-mini | 92.0 | 64.3 | 73.4 | 65.4 | 64.4 | 65.4 | 92.6 | 73.9 |

By highlighting current models' proactiveness limitations, our work provides meaningful insights for researchers seeking to build more collaborative AI systems. However, promoting proactiveness must be carefully balanced to avoid over-questioning or inefficient behavior. While our benchmark promotes interpretability and safe failure modes (i.e., abstention over hallucination), there is a risk of misuse in adversarial settings if models over-rely on user feedback. We release ProactiveBench to support reproducible and community-driven progress toward more robust and human-aware MLLMs.

# E    LICENSES

All original material presented in this work is intended solely for academic research and not for commercial purposes. Below, we report the licenses of the used datasets and models:

- ROD (Lee et al., 2023): This dataset is released without a license.
- VSOD (Liao et al., 2020): MIT License.
- MVP-N (Wang et al., 2022a): MIT License.

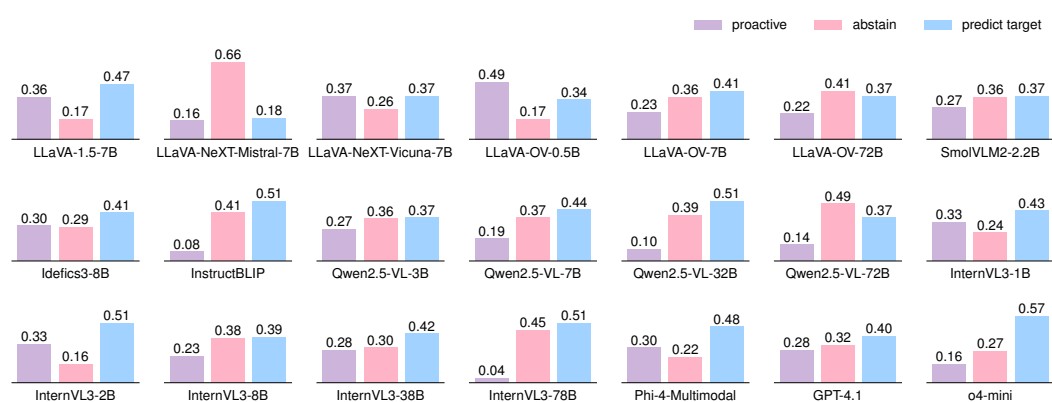

Figure 19: **Action distributions.** We report the action distribution for all evaluated models.

Table 6: **MLLMs results on ProactiveBench with random proactive suggestions.** We report the accuracy (*acc*) in percentages (%) and average number of proactive suggestions (*ps*) for all datasets, with global averages in the last column.

| family | model | ROD acc | ROD ps | VSOD acc | VSOD ps | MVP-N acc | MVP-N ps | IN-C acc | IN-C ps | QD acc | QD ps | CIT acc | CIT ps | COCO acc | COCO ps | avg. acc | avg. ps |
|---|---|---|---|---|---|---|---|---|---|---|---|---|---|---|---|---|---|
| LLaVA-1.5 | 7B | 42.0 | 3.5 | 19.0 | 1.0 | 5.6 | 0.0 | **51.1** | 0.2 | **29.7** | 1.5 | 19.2 | 0.3 | **33.7** | 0.9 | **28.6** | 1.1 |
| LLaVA-NeXT | Mistral-7B | 0.0 | 0.1 | 2.4 | 0.0 | 1.4 | 0.0 | 8.2 | 0.0 | 2.0 | 0.1 | 1.5 | 0.0 | 3.6 | 0.1 | 2.7 | 0.1 |
| | Vicuna-7B | 43.2 | 3.0 | 19.0 | 0.0 | 6.7 | 0.1 | 30.9 | 0.2 | 12.2 | 1.9 | 14.9 | 0.2 | 24.4 | 0.5 | 21.6 | 0.8 |
| LLaVA-OV | 0.5B | 29.5 | 2.0 | 9.5 | 0.0 | 13.7 | 0.5 | 29.1 | 0.4 | 26.0 | 1.0 | 13.6 | 0.4 | 28.7 | 0.6 | 21.5 | 0.7 |
| | 7B | 1.1 | 0.0 | 26.2 | 0.7 | 5.4 | 0.0 | 17.5 | 0.0 | 15.5 | 0.0 | 3.5 | 0.1 | 4.3 | 0.0 | 10.5 | 0.1 |
| SmolVLM2 | 2.2B | 12.5 | 0.8 | 14.3 | 0.0 | 9.3 | 0.0 | 6.3 | 0.0 | 4.1 | 0.0 | 11.1 | 0.5 | 10.4 | 0.0 | 9.7 | 0.2 |
| Idefics3 | 8B | 19.3 | 0.5 | 26.2 | 2.6 | 6.4 | 0.0 | 25.2 | 0.0 | 13.5 | 0.5 | 9.1 | 0.2 | 12.2 | 0.2 | 16.0 | 0.6 |
| InstructBLIP | 7B | 0.0 | 0.7 | 16.7 | 1.2 | 2.0 | 0.6 | 30.2 | 0.1 | 15.5 | 0.6 | **19.7** | 0.2 | 14.3 | 0.4 | 14.1 | 0.5 |
| Qwen2.5-VL | 3B | 6.8 | 0.2 | 9.5 | 0.0 | 5.2 | 0.0 | 27.0 | 0.0 | 8.3 | 0.0 | 5.6 | 0.0 | 8.8 | 0.0 | 10.2 | 0.0 |
| | 7B | 2.3 | 0.0 | 19.0 | 0.0 | 4.9 | 0.0 | 30.3 | 0.0 | 12.4 | 0.0 | 8.3 | 0.0 | 7.1 | 0.0 | 12.1 | 0.0 |
| InternVL3 | 1B | **44.3** | 1.4 | 14.3 | 0.0 | 14.5 | 0.1 | 35.4 | 0.0 | 12.6 | 0.3 | 14.9 | 0.1 | 31.7 | 0.4 | 24.0 | 0.3 |
| | 2B | 17.0 | 0.3 | **28.6** | 0.0 | **20.8** | 0.0 | 43.3 | 0.1 | 14.7 | 0.1 | 18.2 | 0.2 | 27.5 | 0.1 | 24.3 | 0.1 |
| | 8B | 2.3 | 0.0 | 21.4 | 0.9 | 6.8 | 0.0 | 25.8 | 0.0 | 12.7 | 0.0 | 6.8 | 0.1 | 8.7 | 0.0 | 12.1 | 0.2 |
| Phi-4-Multimodal | 6B | 12.5 | 0.6 | 11.9 | 0.2 | 18.6 | 0.0 | 18.6 | 0.1 | 20.1 | 0.1 | 17.2 | 0.2 | 31.2 | 0.6 | 18.6 | 0.3 |

- ImageNet-C (Hendrycks & Dietterich, 2019): Apache License 2.0.

- QuickDraw (Jongejan et al., 2016): CC-BY-4.0.

- ChangeIt (Souček et al., 2022): MIT License.

- MS-COCO (Lin et al., 2014): CC-BY-4.0.

- LLaVA-1.5 (Liu et al., 2024a): Llama2.

- LLaVA-NeXT Vicuna (Liu et al., 2024a): Llama2.

- LLaVA-NeXT Mistral (Liu et al., 2024a): Apache License 2.0.

- LLaVA-OV (Li et al., 2025): Apache License 2.0.

- Qwen2.5-VL (Bai et al., 2025): Apache License 2.0.

- SmolVLM2 (Marafioti et al., 2025): Apache License 2.0.

- Idefics3 (Laurençon et al., 2024): Apache License 2.0.

- InternVL3 (Zhu et al., 2025): Apache License 2.0.

- InstructBLIP (Dai et al., 2023): Llama2.

- Phi-4-Multimodal (Abouelenin et al., 2025): MIT License.

# F  DATASET EXAMPLES

Figures 22 to 28 report dataset examples returned by the environment in the first state.

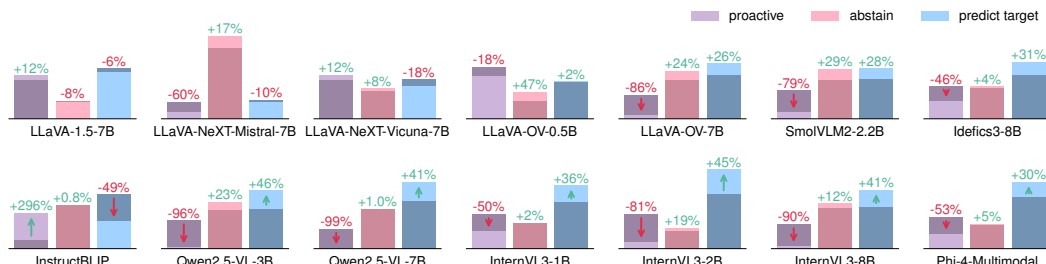

Figure 20: **Action distributions with random proactive options.** Lighter bars describe variations using random proactive suggestions for all evaluated models.

Table 7: **MLLMs results on ProactiveBench by hinting at proactive suggestions.** We report the accuracy (*acc*) in percentages (%) and average number of proactive suggestions (*ps*) for all datasets, with global averages in the last column.

| family | model | ROD acc | ps | VSOD acc | ps | MVP-N acc | ps | IN-C acc | ps | QD acc | ps | CIT acc | ps | COCO acc | ps | avg. acc | ps |
|--------|-------|-----|-----|------|-----|------|-----|------|-----|-----|-----|-----|-----|------|-----|-----|-----|
| LLaVA-1.5 | 7B | 47.7 | 4.9 | 28.6 | 25.9 | 14.7 | 2.3 | 11.1 | 1.3 | 37.9 | 1.8 | 48.7 | 2.2 | 41.0 | 1.5 | 32.8 | 5.7 |
| LLaVA-NeXT | Mistral-7B | 2.3 | 3.7 | 0.0 | 4.7 | 2.5 | 2.4 | 14.3 | 1.1 | 4.8 | 1.0 | 5.8 | 1.1 | 11.3 | 0.5 | 5.9 | 2.1 |
| | Vicuna-7B | 44.3 | 4.9 | 14.3 | 43.3 | 14.0 | 2.3 | 17.8 | 1.2 | 10.4 | 2.3 | 50.8 | 3.4 | 46.4 | 1.3 | 28.3 | 8.4 |
| LLaVA-OV | 0.5B | 44.3 | 5.6 | 9.5 | 29.4 | 17.7 | 1.0 | 19.1 | 1.9 | 36.5 | 2.1 | 44.2 | 6.5 | 38.9 | 1.1 | 30.0 | 6.8 |
| | 7B | 20.5 | 0.6 | 23.8 | 0.7 | 27.7 | 1.0 | 40.6 | 2.1 | 28.5 | 0.5 | 8.8 | 0.5 | 9.2 | 0.2 | 22.7 | 0.8 |
| | 72B | 0.0 | 0.0 | 14.3 | 0.1 | 19.6 | 0.7 | 41.2 | 2.0 | 20.1 | 0.6 | 14.4 | 0.5 | 14.4 | 0.3 | 17.7 | 0.6 |
| SmolVLM2 | 2.2B | 0.0 | 0.1 | 14.3 | 0.2 | 16.1 | 0.5 | 29.2 | 2.2 | 11.2 | 0.8 | 51.3 | 1.8 | 7.8 | 0.1 | 18.6 | 0.8 |
| Idefics3 | 8B | 29.5 | 9.4 | 28.6 | 37.3 | 13.7 | 0.7 | 33.2 | 0.9 | 15.9 | 1.4 | 24.7 | 0.9 | 34.4 | 1.0 | 25.7 | 7.4 |
| InstructBLIP | 7B | 1.1 | 0.5 | 16.7 | 4.9 | 7.4 | 0.1 | 7.9 | 0.1 | 14.2 | 0.1 | 22.7 | 0.2 | 10.0 | 0.0 | 11.4 | 0.8 |
| Qwen-2.5-VL | 3B | 48.9 | 1.3 | 33.3 | 4.2 | 12.6 | 0.5 | 33.8 | 2.6 | 11.1 | 0.5 | 11.9 | 0.6 | 10.5 | 0.1 | 23.1 | 1.4 |
| | 7B | 0.0 | 0.0 | 9.5 | 0.1 | 12.6 | 0.3 | 50.0 | 2.1 | 23.8 | 0.9 | 6.3 | 0.2 | 6.3 | 0.0 | 15.5 | 0.5 |
| | 32B | 10.2 | 0.8 | 2.4 | 0.2 | 25.4 | 1.2 | 40.9 | 1.4 | 26.4 | 1.1 | 15.7 | 0.9 | 24.1 | 0.6 | 20.7 | 0.9 |
| | 72B | 0.0 | 0.0 | 9.5 | 0.6 | 28.2 | 1.3 | 44.7 | 2.4 | 26.8 | 1.9 | 23.7 | 1.1 | 32.1 | 0.9 | 23.6 | 1.2 |
| InternVL3 | 1B | **62.5** | 2.9 | 23.8 | 1.4 | 33.0 | 1.6 | 26.4 | 2.0 | 25.7 | 2.3 | 39.9 | 2.1 | 31.0 | 0.7 | 34.6 | 1.9 |
| | 2B | 42.0 | 1.1 | 52.4 | 14.4 | 34.4 | 1.1 | 41.3 | 1.5 | 29.2 | 1.8 | 32.8 | 2.5 | 52.8 | 0.9 | 40.7 | 3.3 |
| | 8B | 2.3 | 0.0 | 19.0 | 0.1 | 19.0 | 0.6 | 38.6 | 1.1 | 26.7 | 1.1 | 9.6 | 0.3 | 13.7 | 0.2 | 18.4 | 0.5 |
| | 38B | 3.4 | 0.0 | 33.3 | 2.0 | 38.2 | 1.3 | 53.5 | 1.9 | 35.7 | 1.8 | 29.8 | 1.3 | 56.0 | 0.9 | 35.7 | 1.3 |
| | 78B | 0.0 | 0.0 | 31.0 | 1.4 | 18.2 | 0.3 | 57.0 | 0.8 | 11.4 | 0.4 | 29.0 | 1.3 | 23.9 | 0.2 | 24.3 | 0.6 |
| Phi-4-Multimodal | 6B | 9.1 | 0.4 | 9.5 | 1.1 | 21.4 | 0.2 | 35.1 | 2.3 | 34.3 | 1.3 | 23.2 | 0.5 | 26.0 | 0.4 | 22.7 | 0.9 |
| OpenAI | GPT-4.1 | 0.0 | 0.0 | 7.1 | 0.8 | **52.2** | 2.8 | 30.8 | 3.0 | 33.0 | 2.5 | 24.7 | 0.7 | 92.8 | 0.1 | 34.4 | 1.4 |
| | o4-mini | 20.5 | 0.2 | **54.8** | 4.6 | 31.4 | 0.5 | **66.8** | 0.8 | **62.0** | 1.3 | **59.8** | 1.8 | **94.6** | 0.1 | **55.7** | 1.3 |

# G   LLM USAGE DECLARATION

During the writing of this paper, we used LLMs for polishing writing and proofreading the manuscript.

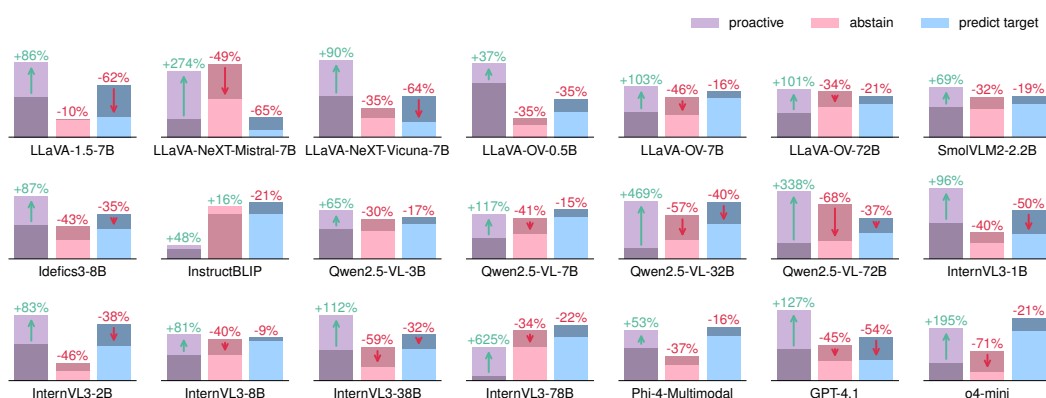

Figure 21: **Action distributions with hints.** Bars describe action distributions with (light) and without (dark) hints in the prompt for all evaluated models.

Table 8: **MLLMs results on ProactiveBench by conditioning on conversation histories.** We report the accuracy (*acc*) in percentages (%) and average number of proactive suggestions (*ps*) for all datasets, with global averages in the last column. We omit models not supporting multi-image inference.

| family | model | ROD | | VSOD | | MVP-N | | IN-C | | QD | | CIT | | COCO | | avg. | |
|---|---|---|---|---|---|---|---|---|---|---|---|---|---|---|---|---|---|
| | | acc | ps | acc | ps | acc | ps | acc | ps | acc | ps | acc | ps | acc | ps | acc | ps |
| LLaVA-OV | 0.5B | 1.1 | 6.0 | 9.5 | 10.9 | 8.5 | 0.4 | 5.4 | 1.6 | 15.2 | 1.8 | 17.7 | 6.8 | 16.9 | 0.7 | 10.6 | 4.0 |
| | 7B | 0.0 | 0.0 | 14.3 | 5.8 | 6.4 | 0.1 | 10.7 | 1.3 | **22.9** | 0.7 | 12.4 | 1.4 | 3.5 | 0.1 | 10.0 | 1.3 |
| SmolVLM2 | 2.2B | 0.0 | 0.0 | 7.1 | 0.4 | 9.9 | 0.1 | 4.5 | 0.9 | 4.8 | 0.4 | 20.7 | 1.0 | 5.5 | 0.0 | 7.5 | 0.4 |
| Idefics3 | 8B | **13.6** | 1.3 | 11.9 | 3.8 | 6.3 | 0.1 | **24.4** | 1.5 | 9.5 | 0.7 | 18.7 | 0.9 | 8.7 | 0.2 | 13.3 | 1.2 |
| Qwen2.5-VL | 3B | 0.0 | 0.0 | 11.9 | 0.0 | 5.0 | 0.0 | 14.2 | 2.5 | 8.0 | 0.3 | 14.1 | 1.0 | 6.4 | 0.0 | 8.5 | 0.6 |
| | 7B | 0.0 | 0.0 | 0.0 | 0.0 | 4.4 | 0.0 | 24.3 | 1.8 | 8.3 | 0.2 | 9.1 | 0.3 | 4.9 | 0.0 | 7.3 | 0.3 |
| InternVL3 | 1B | 0.0 | 6.4 | 16.7 | 8.1 | 11.7 | 0.6 | 11.5 | 1.7 | 7.5 | 0.8 | 10.6 | 2.6 | 15.2 | 0.2 | 10.4 | 2.9 |
| | 2B | 0.0 | 0.1 | **28.6** | 6.3 | 16.4 | 0.2 | 5.9 | 2.2 | 8.6 | 0.9 | 14.4 | 4.2 | **30.1** | 0.3 | **14.8** | 2.0 |
| | 8B | 0.0 | 0.0 | 9.5 | 7.5 | 5.7 | 0.1 | 9.2 | 1.3 | 6.9 | 1.0 | 4.5 | 1.9 | 6.9 | 0.0 | 6.1 | 1.7 |
| Phi-4-Multimodal | 6B | 0.0 | 0.0 | 7.1 | 12.7 | **18.9** | 0.0 | 8.6 | 2.0 | 14.9 | 0.8 | **26.3** | 6.5 | 15.9 | 0.4 | 13.1 | 3.2 |

Table 9: **MLLMs results on ProactiveBench by conditioning on few-shots.** We report the accuracy (*acc*) in percentages (%) and average number of proactive suggestions (*ps*) for all datasets, with global averages in the last column. We omit models not supporting multi-image inference.

| family | model | 1 sample | | | | 3 samples | | | |
|---|---|---|---|---|---|---|---|---|---|
| | | ROD | | MVP-N | | ROD | | MVP-N | |
| | | acc | ps | acc | ps | acc | ps | acc | ps |
| LLaVA-OV | 0.5B | 21.6 | 7.4 | 12.8 | 0.6 | 21.6 | 6.6 | 13.2 | 0.6 |
| | 7B | 9.1 | 0.2 | **17.9** | 0.7 | 19.3 | 0.4 | 20.0 | 0.9 |
| SmolVLM2 | 2.2B | 0.0 | 0.1 | 7.7 | 0.2 | 0.0 | 0.1 | 0.4 | 0.0 |
| Idefics3 | 8B | **31.8** | 1.3 | 8.6 | 0.1 | 34.1 | 1.2 | 10.0 | 0.4 |
| Qwen2.5-VL | 3B | 2.3 | 0.2 | 8.0 | 0.1 | 2.3 | 0.0 | 9.1 | 0.3 |
| | 7B | 0.0 | 0.0 | 7.9 | 0.1 | 4.5 | 0.1 | **20.2** | 0.6 |
| InternVL3 | 1B | 0.0 | 9.8 | 12.2 | 0.7 | 0.0 | 9.5 | 13.4 | 0.8 |
| | 2B | 0.0 | 0.7 | 17.3 | 0.5 | 0.0 | 0.6 | 16.4 | 0.5 |
| | 8B | 0.0 | 0.0 | 6.2 | 0.1 | 0.0 | 0.0 | 6.4 | 0.1 |
| Phi-4-Multimodal | 6B | 2.3 | 0.2 | 17.0 | 0.1 | **38.6** | 3.7 | 12.7 | 1.5 |

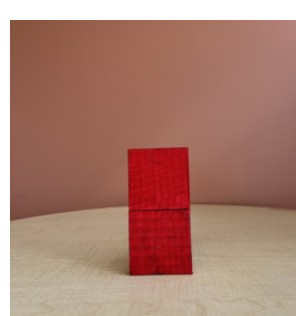

```
What missing details are obscured by the red blocks?
Choose from the following options. Options:
A. Move the red blocks to the left.
B. Move the red blocks to the right.
C. I cannot tell.
D. Screwdriver.
E. Banana.
F. Vase.
G. Spatula.
Please only return one of the options without any other words.
```

Figure 22: **ROD input example**. In the first step, the ROD environment returns images of completely occluded target objects.

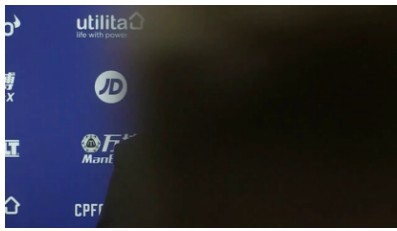

```
This is a frame extracted from a video. Answer the following
question.
Can you say who is speaking?
Choose from the following options. Options:
A. I cannot answer this question.
B. Rewind the video.
C. Wait for the occlusion to disappear.
D. Monika schnitzer.
E. Ursula von der leyen.
F. Ge you.
G. José mourinho.
Please only return one of the options without any other words.
```

Figure 23: **VSOD input example**. In the first step, the VSOD environment returns video frames of occluded subjects.

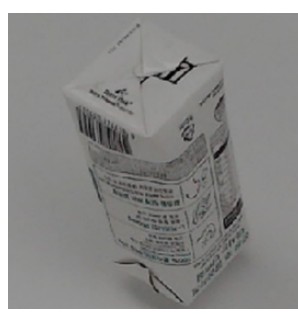

```
Could you name the object in this image?
Choose from the following options. Options:
A. Change the camera angle of the object.
B. I do not know what is this object.
C. Selex whey protein drink peach.
D. Selex sports whey protein powder peach.
E. Selex sports whey protein powder chocolate.
F. Selex whey protein drink chocolate.
Please only return one of the options without any other words.
```

Figure 24: **MVP-N input example**. In the first step, the MVP-N environment returns uninformative object views.

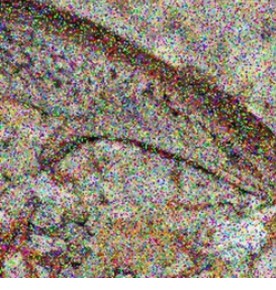

```
Provide the classification of the object in the image.
Choose from the following options. Options:
A. Denoise the image.
B. I do not know what is this object.
C. Increase image resolution.
D. Reduce brightness.
E. Deblur the image.
F. Perfume.
G. Great_pyrenees.
H. Alligator_lizard.
I. Cello.
Please only return one of the options without any other words.
```

Figure 25: **ImageNet-C input example**. In the first step, the IN-C environment returns heavily corrupted images.

```
Describe the object in the quickdraw in terms of its category.
Choose from the following options. Options:
A. I cannot answer this question.
B. Make this drawing more complete.
C. The eiffel tower.
D. Potato.
E. Bed.
F. Tooth.
Please only return one of the options without any other words.
```

Figure 26: **QuickDraw input example**. In the first step, the QD environment returns the first stroke of a sketch.

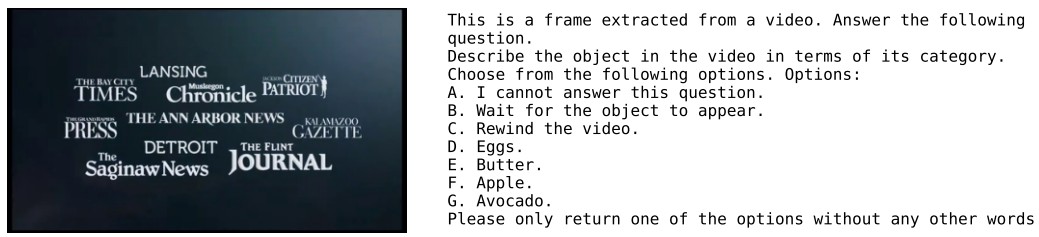

```
This is a frame extracted from a video. Answer the following
question.
Describe the object in the video in terms of its category.
Choose from the following options. Options:
A. I cannot answer this question.
B. Wait for the object to appear.
C. Rewind the video.
D. Eggs.
E. Butter.
F. Apple.
G. Avocado.
Please only return one of the options without any other words.
```

Figure 27: **ChangeIt input example**. In the first step, the CIT environment returns video frames where the target object or action will appear in the future.

```
Identify the object in this image.
Choose from the following options. Options:
A. Move the camera to the left.
B. I cannot answer this question.
C. Move the camera down.
D. Bowl.
E. Sink.
F. Cup.
G. Toilet.
Please only return one of the options without any other words.
```

Figure 28: **MS-COCO input example**. In the first step, the COCO environment returns images where object details are removed.

