# OpenReview forum: "ProactiveBench: Benchmarking Proactiveness in Multimodal Large Language Models"
_ICLR.cc/2026/Conference — ICLR 2026 Conference Withdrawn Submission_

### Official Review · Reviewer_cWmi · 2025-10-28

**Soundness:** 2
**Presentation:** 3
**Contribution:** 2
**Rating:** 2
**Confidence:** 3

**Summary:**

The paper presents a benchmark to measure the "proactiveness" of MLLMs, that is, when the information in the image is insufficient to answer a question, can the model ask for more information by requesting a different view of the object in the image? This can include rewinding/forwarding through a video, deblurring an image, moving the camera left/right, etc. The authors show that all MLLMs available have a low degree of "proactiveness" and techniques like hinting in prompt, ICL, and including conversation histories is insufficient to improve proactiveness.

**Strengths:**

The key idea of measuring "proactiveness" is quite interesting, since it is generally assumed that the MLLM is passively receiving the input with no control over it. The paradigm where the MLLM can change its own input is quite novel, in my opinion.

**Weaknesses:**

1. The ps rate of many models, especially frontier models such as GPT 4.1 and o4-mini is unreasonably low. The hint and prompt seem inadequate for eliciting proactiveness, and the results seem misleading. I would suggest the following:

(a) Explicitly tell the model (in system prompt if available) that it can control the input, for example, it can control the camera to move left/right.

(b) Also, give the model a tool that can do this, instead of simply including it as an option in a multiple-choice question.

 (c) Use CoT evaluation where the model can call the tool in its chain-of-thought.

Both frontier models (GPT 4/5, Gemini 2.5) and the latest open-source models (like Qwen3 which have been trained on tool calling) can be evaluated in this setting.

2. A more thorough analysis of frontier models would very much improve the work, as many of the older open-source models were not trained with tool calling or agentic applications in mind.

3. Reporting the number of proactive actions and distribution over actions is not very insightful. It is more interesting to give the models a budget of number of proactive actions to answer the question, and then measure things like over-confidence or under-confidence. Over-confidence could be defined as the % of samples where the model answers incorrectly when it could have answered correctly after a few proactive suggestions. Under-confidence would be the inverse of this.

**Questions:**

Please address the weaknesses above

---

### Official Review · Reviewer_W4fg · 2025-10-29

**Soundness:** 3
**Presentation:** 2
**Contribution:** 3
**Rating:** 4
**Confidence:** 4

**Summary:**

This paper introduces ProactiveBench, a multimodal benchmark designed to evaluate an MLLM's ability to proactively request additional information when the current visual context is insufficient to answer a question. The benchmark covers scenarios of spatial and temporal occlusions, uninformative views, low-quality or corrupted images, missing visual details, temporal ambiguities, and camera movement. It is constructed by repurposing seven existing datasets, and the evaluation metrics include final accuracy and the average number of proactive suggestions (ps).

Through extensive evaluation across multiple MLLMs, the study finds that current models remain largely reactive rather than proactive, and that proactiveness does not naturally emerge in existing MLLMs. Overall, the benchmark highlights a challenging and underexplored problem that future multimodal models need to address.

**Strengths:**

- Clearly defines and systematically evaluates proactiveness, the ability of MLLMs to actively request additional information when the current visual context is ambiguous or incomplete.
- Covers seven diverse and realistic interaction scenarios, including occlusion, viewpoint change, low image quality, incomplete sketches, temporal ambiguity, and camera movement.
- Conducts a large-scale evaluation across 21 MLLMs, reporting both accuracy (acc) and average proactive suggestions (ps), together with an oracle reference, which provides a meaningful diagnostic gap between reactive and proactive behaviors.
- Explores the influence of hints, conversation history, and few-shot conditioning, giving a thorough analysis of when and why models fail to exhibit proactiveness.
- Includes a random-invalid-suggestion control to differentiate genuine proactive reasoning from mere guessing or low abstention tendency.
- Fills a clear gap in existing multimodal benchmarks, which mainly focus on perception or reasoning but overlook active information-seeking behavior.
- Demonstrates that proactiveness is not an emergent property of current MLLMs, providing concrete evidence and a new direction for model alignment and evaluation research.

**Weaknesses:**

- **Evaluation format limitation** The main experiments are conducted in a multiple-choice setting with predefined options, which can be relatively constrained and semantically suggestive. This limits the diversity of potential actions and may inflate apparent proactiveness. A richer pool of distractors would better stress-test proactive behavior.

- **Lack of statistical significance reporting** Figures and tables visualize average accuracy and proactive-suggestion rates across models/interventions, but do not report variance, confidence intervals, or significance tests; this makes it hard to judge the robustness of observed differences and intervention effects. (e.g., hint/history analyses report mean shifts only.)

- **Limited metric granularity** The benchmark primarily reports **final accuracy (acc)** and **average proactive suggestions (ps)**. However, tasks differ in how many steps are required to reach an informative context, so a single global mean may be misleading. A normalized efficiency metric, e.g., (steps moved) / (steps to reach a sufficient frame) would capture per-task difficulty and model efficiency more faithfully.

- **Under-analyzed side effects of hints and history** Hinting increases the proactive-suggestion rate but also induces blind exploration, while preserving conversation history raises suggestion count yet lowers accuracy. These failure modes are noted but not deeply decomposed. A deeper breakdown, e.g., when repetition starts and which scenarios amplify these side effects, would make the analysis more insightful.

**Questions:**

Please refer to the points discussed in the *Weaknesses* section.

---

### Official Review · Reviewer_3EJZ · 2025-10-31

**Soundness:** 2
**Presentation:** 2
**Contribution:** 2
**Rating:** 4
**Confidence:** 4

**Summary:**

This paper introduces ProactiveBench, a novel benchmark for evaluating proactiveness (the ability to request additional visual cues from users) in MLLMs. The authors repurpose seven existing datasets (ROD, VSOD, MVP-N, ImageNet-C, QuickDraw, ChangeIt, and MS-COCO) to create scenarios requiring human intervention, such as moving occluding objects, improving image quality, or requesting more details. They evaluate 21 MLLMs and find that current models generally lack proactiveness, with no clear correlation between model capacity and proactive behavior.

**Strengths:**

- Quality: The experimental methodology is sound, with comprehensive evaluation across 21 models and seven scenarios.

- Clarity: Good presentation with clear visualizations of benchmark scenarios.

- Significance: The benchmark provides a concrete framework for evaluating an important capability, proactiveness, from existing datasets.

**Weaknesses:**

1. Literature gap:
  - The paper fails to adequately mention existing work on evaluating and enhancing active perception and interactive VQA. Similar concepts have been explored under different terminology. [1][2][3][4].
  - If author believe that your addressed proactiveness are totally different from existing concpet, active vision or active perception, please justify it with sufficient literal support from both computer science and cognition fields.

2. Benchmark accessibility concern: While not explicitly stated in the paper, the reliance on proprietary models (like GPT-4o) for evaluation, which is common in such benchmarks, creates significant accessibility barriers. This increases research costs and limits reproducibility, making the benchmark less practical for widespread adoption by the research community. This is a substantial practical limitation that affects the benchmark's utility.

3. Overstated novelty: Claims of introducing the "first" benchmark for this capability are inaccurate given prior work.

4. Limited analysis of root causes: Minimal insight into why models struggle with proactiveness.

5. Insufficient discussion of solutions and insights:
  - Little guidance on improving proactiveness.
  - Some models present all-zero results, show that these models do not possess corresponding ability, and are equally bad. What else can be derived?
  - Generally speaking, larger scale models are better than smaller models. However, results in this paper shows that 0.5B and 1B models outperform all the other models in ROD task in table 2. This is surprising and attractive findings, but no further analysis are provided.

[1] MLLMs Know Where to Look: Training-free Perception of Small Visual Details with Multimodal LLMs, ICLR 2025
[2] ActiView: Evaluating Active Perception Ability for Multimodal Large Language Models, ACL 2025
[3] Iqa: Visual question answering in interactive environments, CVPR 2018
[4] Bajcsy, R. (1988). Active perception. Proceedings of the IEEE, 76(8), 966-1005

**Questions:**

Additional question:

1. How do you differentiate "proactiveness" from previously studied concepts like "interactive VQA", "adaptive vision" or "clarification-requesting"? How does ProactiveBench fundamentally differ from existing benchmarks for interactive vision, active vision, and active perception? What specific aspects of proactiveness does your benchmark capture that previous work missed?

2. Have you compared your results to models specifically designed for interactive/adaptive/active vision?

---

### Official Review · Reviewer_yk1G · 2025-11-01

**Soundness:** 4
**Presentation:** 3
**Contribution:** 3
**Rating:** 6
**Confidence:** 3

**Summary:**

This paper introduces ProactiveBench, a novel benchmark designed to evaluate the proactiveness of Multimodal Large Language Models (MLLMs)—that is, their ability to ask for additional visual information when faced with an ambiguous or unanswerable query, rather than hallucinating or abstaining. The benchmark is constructed by repurposing seven existing datasets (ROD, VSOD, MVP-N, ImageNet-C, QuickDraw, ChangeIt, MS-COCO) into a multi-turn, multiple-choice evaluation framework across seven distinct scenarios involving occlusions, poor image quality, uninformative views, and temporal ambiguities.

**Strengths:**

- The concept of "proactiveness" as a distinct capability is novel and well-motivated for interactive AI systems. The formalization as seeking additional information rather than guessing or refusing is an important research direction.
- The benchmark covers diverse visual challenges across seven distinct scenarios. Comprehensive evaluation across 21 models including both open and closed-source systems.
- The paper is generally well-written with clear motivation and problem definition.

**Weaknesses:**

- Aggressive and insufficiently justified filtering methodology:
The 25% threshold removes 58% of samples (from 17,909 to 7,557), which is extremely aggressive. The paper states this focuses evaluation on "proactive behaviors" but doesn't justify why 25% was chosen over 15%, 30%, or other thresholds. This filtering may create an unrepresentatively difficult benchmark that doesn't reflect real-world distributions where some ambiguous queries are answerable. The unfiltered results (Appendix, Table 4) show 46% average accuracy vs. 17.5% filtered—this dramatic difference suggests the benchmark may be testing extreme cases rather than typical proactiveness scenarios.
- Multiple-choice constraint: While justified for tractability, this significantly constrains how proactiveness can be expressed. Real proactive behavior might involve asking clarifying questions, requesting specific types of information, or explaining uncertainty—none of which are captured. The free-form results in Appendix A are interesting but relegated to secondary status and limited to 6 models.
- Benchmark construction and quality concerns: Repurposing vs. purpose-building: All seven datasets were designed for other tasks. While clever, this raises questions about whether they truly capture proactiveness or just task-specific patterns. Purpose-built scenarios might better isolate the target capability. Annotation quality: VSOD required manual annotation of public figures and event types—no inter-annotator agreement or quality metrics are reported. The 55.3% of ImageNet-C samples that were "too easy" suggests significant annotation noise. Limited scenario diversity: All scenarios involve visual ambiguity. What about proactiveness for knowledge gaps, logical contradictions, or ambiguous language? The benchmark's scope is narrower than the concept warrants.

**Questions:**

- Why do models abstain? The paper shows that apparent proactiveness correlates with lower abstention but doesn't investigate why some models abstain more. Is it instruction-tuning, model architecture, or training data?
- No error taxonomy: Beyond aggregate statistics, there's no categorization of failure modes. When models predict incorrectly, is it because they attempted proactiveness but chose wrong actions, or because they guessed categories too early?
- Scenario difficulty analysis: Why is ROD (8.2% accuracy) so much harder than ImageNet-C (33.7%)? Understanding this could inform benchmark design and model development.
- Random proactive options (Section 4.3): The interpretation is unclear. When models choose "rewind the video" for QuickDraw, does this indicate they don't understand the task, or simply that they prefer any action over abstention? The experiment conflates task understanding with action preference.
- Some dataset statistics (e.g., "~230 images/sample" for VSOD) use approximations where exact numbers would be better.

---

### Note · Authors · 2025-11-12

I have read and agree with the venue's withdrawal policy on behalf of myself and my co-authors.